# A loop region of BAFF controls B cell survival and regulates recognition by different inhibitors

Michele Vigolo[1], Melissa G. Chambers[2], Laure Willen[1], Dehlia Chevalley[1], Klaus Maskos[3], Alfred Lammens[3], Aubry Tardivel[1], Dolon Das[1], Christine Kowalczyk-Quintas[1], Sonia Schuepbach-Mallepell[1], Cristian R. Smulski[1], Mahya Eslami[1], Antonius Rolink[4], Edith Hummler[5], Eileen Samy[6], Yves Fomekong Nanfack[6], Fabienne Mackay[7], Maofu Liao[2], Henry Hess[8], Xuliang Jiang[6] & Pascal Schneider [ID] [1]

The B cell survival factor (TNFSF13B/BAFF) is often elevated in autoimmune diseases and is targeted in the clinic for the treatment of systemic lupus erythematosus. BAFF contains a loop region designated the flap, which is dispensable for receptor binding. Here we show that the flap of BAFF has two functions. In addition to facilitating the formation of a highly active BAFF 60-mer as shown previously, it also converts binding of BAFF to TNFRSF13C (BAFFR) into a signaling event via oligomerization of individual BAFF-BAFFR complexes. Binding and activation of BAFFR can therefore be targeted independently to inhibit or activate the function of BAFF. Moreover, structural analyses suggest that the flap of BAFF 60-mer temporarily prevents binding of an anti-BAFF antibody (belimumab) but not of a decoy receptor (atacicept). The observed differences in profiles of BAFF inhibition may confer distinct biological and clinical efficacies to these therapeutically relevant inhibitors.

---

[1] Department of Biochemistry, University of Lausanne, 1066 Epalinges, Switzerland. [2] Department of Cell Biology, Harvard Medical School, Boston, MA 02115 USA. [3] Proteros Biostructures GmbH, 82152 Planegg, Germany. [4] Department of Biomedicine, University of Basel, 4058 Basel, Switzerland. [5] Department of Pharmacology and Toxicology, University of Lausanne, 1011 Lausanne, Switzerland. [6] EMD Serono Research & Development Institute, Billerica, MA 01821, USA. [7] Department of Immunology, Monash University, Melbourne, VIC 3004, Australia. [8] Merck KGaA, 64293 Darmstadt, Germany. Deceased: Antonius Rolink. Correspondence and requests for materials should be addressed to P.S. (email: pascal.schneider@unil.ch)

B cells actively participate in the adaptive immune response. Their main function is to produce antibodies that protect against bacterial infections. Antibodies are respectively absent or low in patients with X-linked agammaglobulinemia, who selectively lack B but not T cells, and in patients with common variable immunodeficiency. In both cases, infections of the respiratory and gastro-intestinal tracts are the most common symptoms that can be largely prevented by transfer of immunoglobulins[1,2]. Systemic lupus erythematosus (SLE), on the contrary, is characterized by excessive B cell activity and production of autoantibodies that form autoimmune complexes, trigger complement activation, and deposit in glomeruli which can cause nephropathies[3]. The B cell activation factor of the tumor necrosis factor (TNF) family (BAFF, also known as TNFSF13B or B lymphocyte stimulator, BLyS) is often elevated in SLE (reviewed in refs. [4,5]). An anti-BAFF therapy (belimumab, trade name Benlysta) was approved in 2011 for the treatment of adult patients with active, autoantibody-positive SLE. Other BAFF inhibitors are in clinical development, some of which, like a TACI (transmembrane activator and calcium modulator and cyclophilin ligand interactor, TNFRSF13B)-Fc decoy receptor (atacicept), additionally inhibit a proliferation-inducing ligand (APRIL, also known as TNFSF13) (reviewed in refs. [4,5]). BAFF and APRIL are important fitness and survival factors for mature B cells and plasma cells[6]. They are homo-trimeric type-II transmembrane proteins that can be proteolytically processed at furin consensus cleavage sites to release soluble cytokines[7–9]. BAFF is expressed by cells of myeloid origin and by stromal cells[10]. It binds to three receptors, BAFF receptor (BAFFR, TNFRSF13C), TACI, and B cell maturation antigen (BCMA, TNFRSF17), while APRIL interacts only with TACI and BCMA (reviewed in ref. [6]). While BAFFR, TACI, and BCMA are all expressed in B cells at different stages of development, BAFFR is the first one to be expressed and the only one required for survival of transitional and mature naive B cells[11,12]. TACI is expressed in B cells upon activation[13] and is expressed at higher levels in marginal zone B cells[14] while expression of BCMA may require down-regulation of BAFFR[15] and is found in germinal center B cells[16] and in terminally differentiated B cells[17,18].

Soluble BAFF 3-mers can exist as such, or further assemble, at least for human BAFF in vitro, into ordered dodecahedrons called BAFF 60-mer[19]. Primary mouse B cells activated in vitro with an anti-B cell receptor antibody can receive survival signals through either BAFFR or TACI. In this system, BAFFR responds to all forms of BAFF, while TACI is only activated by higher order multimers of BAFF or APRIL[20], suggesting that soluble BAFF 3-mer provides the general survival signal for B cells, while other forms of BAFF and APRIL, such as BAFF 60-mer, proteoglycan-bound APRIL, or the membrane-bound ligands, would serve distinct or additional functions. This view fits with the observation that mice expressing uncleavable BAFF display reduced levels of soluble BAFF and a phenotype similar to that of Baff (Tnfsf13b)$^{-/-}$ mice[21]. In order to form BAFF 60-mer, 3-mers interact with their neighbors via a loop region of 10 amino acid residues located between β-sheets D and E of BAFF, called as the flap[19]. In this study, we describe that this flap is dispensable for receptor binding but essential for the activity of soluble BAFF 3-mers acting on BAFFR. This function of the flap beyond 60-mer assembly is an essential feature of the mechanism of action of BAFF and suggests new strategies to either activate or inhibit BAFF activity. We also describe that the BAFF inhibitors belimumab and atacicept differ in their mechanism of BAFF 60-mer inhibition.

## Results

### Flap mutations do not affect binding of BAFF to receptors. To address the question of how BAFF activates BAFFR, and therefore

promotes B cell survival, mutations were designed to inhibit interactions between adjacent flaps. Mutation of histidine 218 (H218A; in mouse: H242A) at the periphery of the flap prevents formation of BAFF 60-mer[22]. Mutation E223K (in mouse: E247K) was also generated because glutamic acid 223 makes a hydrogen bond with lysine 216 at the center of the flap–flap interaction (Fig. 1a).

Recombinant wild type (WT) and flap mutant BAFF, FLAG-tagged or naturally cleaved, were size-fractionated to recover 3-mers, quantified by western blot (Fig. 1b, c) and found to all bind immobilized recombinant receptors (BAFFR, TACI, and BCMA) similarly (Fig. 1d). This result was confirmed in a competitive enzyme-linked immunosorbent assay (ELISA) where FLAG-tagged WT or mutant BAFF competed with naturally cleaved untagged WT BAFF for receptor binding. Competition curves were indistinguishable, indicating no major difference in the binding affinities of WT and mutant BAFF to BAFFR (Fig. 1e). WT and mutant BAFF bound identically to endogenous BAFFR on BJAB Burkitt lymphoma cells, and to TACI expressed in BAFFR-ko BJAB cells (Supplementary Fig. 1). We conclude that flap mutations selected in this study do not affect binding of BAFF to recombinant receptors, agreeing with crystal structures of BAFF–BAFFR complexes in which flaps make no contact with receptors[23,24].

**Flap mutations prevent human BAFF to form 60-mers.** Naturally cleaved forms of human BAFF eluted from a size-exclusion chromatography column as 3-mers (fractions 15–16), while mouse BAFF formed larger complexes and eluted in fractions 13–14 (Fig. 2a, b). Only WT human BAFF also eluted as BAFF 60-mer in fractions 9–10 (Fig. 2a). All mouse BAFF constructs produced some high-molecular-weight aggregates eluting in the void volume of the column (fractions 7–9) (Fig. 2b), but in contrast to human BAFF 60-mer, these aggregates were inactive when tested for activity on reporter cells (Fig. 2a, b). Thus, as predicted, both flap mutants prevent 60-mer formation in human BAFF. Lack of active BAFF 60-mer formation in naturally cleaved mouse BAFF may result from an additional exon present in mouse and rat Baff genes that introduces 30 amino acids at the N-terminus of soluble BAFF. This N-terminal extension possibly interferes with 60-mer assembly by steric hindrance (reviewed in ref. [25]).

**Flap–flap interactions are required for BAFF activity.** WT human BAFF 3-mer was active on BAFFR:TNF receptor super-family member 6 (Fas) reporter cells (Fig. 2a). While this activity may be due to re-association into 60-mer, this is unlikely to be the only cause of activity since hBAFF H218A and mBAFF H242A, which cannot form 60-mer, were also active (Fig. 2a, b). Interestingly, hBAFF E223K and mBAFF E247K were inactive when tested on BAFFR:Fas reporter cells (Fig. 2a, b), showing that the two flap mutations tested in this study are not equivalent. This suggests that the flap could serve functions different from that of allowing 60-mer formation.

**B cell lymphopenia in E247K flap mutant knock-in mice.** A knock-in mouse carrying point mutation E247K in exon 6 of the Baff gene was generated to validate the effect of BAFF flap mutation in a physiologically relevant setting (Supplementary Fig. 2). Endogenous serum levels of circulating BAFF were similar in littermates of Baff$^{wt/wt}$ (WT), Baff$^{wt/E247K}$ (heterozygous), and Baff$^{E247K/E247K}$ (knock-in), but were at background in Baff$^{-/-}$ (knock-out) mice (Fig. 3a). The same was true when receptor binding-competent BAFF was monitored using a recombinant receptor (TACI-Fc) and an antibody (Fig. 3b). We conclude that

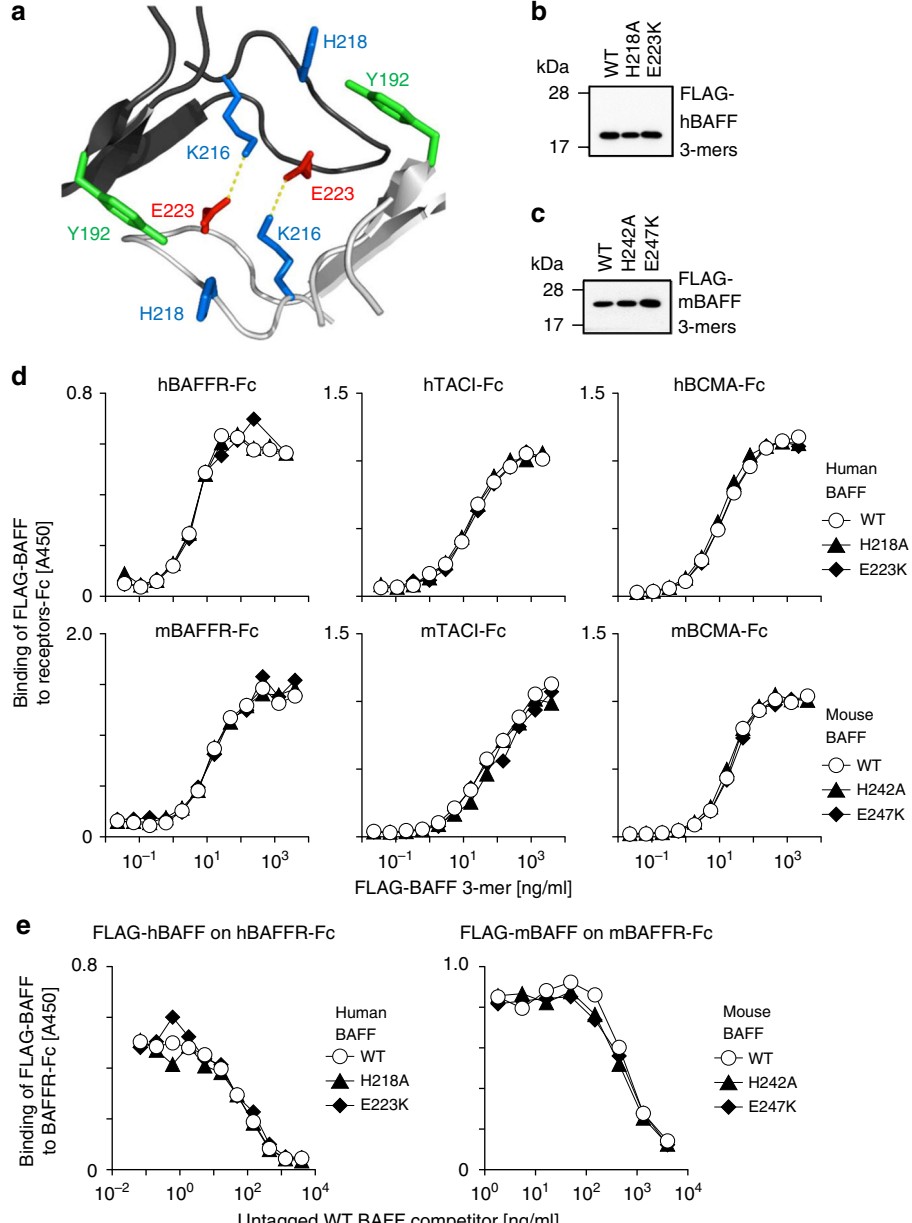

**Fig. 1** Mutations in the flap do not affect binding of BAFF to its receptors. **a** Residues involved in flap–flap interactions between adjacent BAFF 3-mers. Parts of BAFF monomers in adjacent 3-mers are shown in black and white, respectively. Side chains of some residues known or predicted to be important for flap–flap interactions are depicted and numbered according to the human BAFF sequence. The illustration was drawn from coordinates PDB-ID 4V46[23]. **b** FLAG-tagged human BAFF 3-mers wild type (WT) or with mutations (H218A and E223K) in flap residues were produced by transient transfection of 293 T cells, recovered in supernatants, size fractionated by gel filtration and adjusted to comparable concentrations. 80 ng of each protein was analyzed by anti-FLAG western blotting. **c** Same as panel **b**, but for FLAG-tagged mouse BAFF with the corresponding flap mutations. **d** ELISA was used to monitor the binding of WT or mutant FLAG-tagged BAFF to recombinant BAFFR-Fc, TACI-Fc, or BCMA-Fc, with anti-FLAG detection. Binding of human BAFF was probed on human receptors (graphs on the top), and binding of mouse BAFF on mouse receptors (graphs at the bottom). The experiment was performed twice. **e** Titrated amounts of untagged WT BAFF were added to BAFFR-Fc coated in an ELISA plate, followed by a fixed amount of FLAG-BAFF (WT or the indicated mutants) and detection of binding with an anti-FLAG antibody. Absence of signal reveals that untagged BAFF had bound to BAFFR-Fc. Left graph: human BAFF on hBAFFR-Fc. Right graph: mouse BAFF on mBAFFR-Fc. The experiment was performed once for human BAFF and twice for mouse BAFF. See also Supplementary Fig. 1

knock-in mice express and conventionally process BAFF E247K that is then able to bind receptors. Despite the presence of circulating BAFF, the B cell phenotype of knock-in mice resembled that of BAFF knock-out with reduced numbers of CD19+CD93− mature B cells in the spleen, decreased B/T ratio in lymph nodes, but normal numbers of (BAFF-independent) immature CD19+ CD93+ splenic B cells (Fig. 3c–f, Supplementary Fig. 3, Supplementary Table 1). There was a non-significant trend ($p > 0.05$ by

one-way Anova) for slightly higher numbers of mature B cells in knock-in mice compared to knock-out, which might reflect a residual activity of either soluble or membrane-bound BAFF E247K (Fig. 3c–f). These results indicate that BAFF E247K cannot, or only very poorly, activate BAFFR in vivo.

In the absence of B cells, circulating levels of BAFF usually rise[20,26]. We wondered why BAFF levels in B lymphopenic knock-in mice were not higher than in WT and we found that the

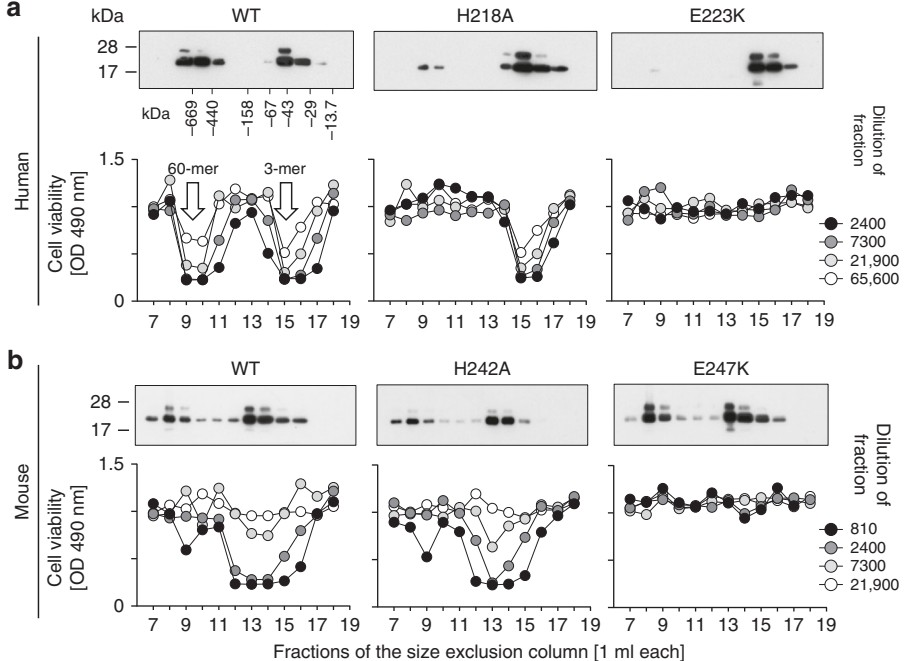

**Fig. 2** Flap mutations affecting 60-mer formation: one of them additionally affects activity of BAFF 3-mer. Naturally cleaved, untagged human or mouse BAFF, with or without the indicated mutations in the flap, were recovered in supernatant of 293 T cells transiently transfected with plasmids encoding the full length wild type (WT) or mutant BAFF. Concentrated supernatants were fractionated by size-exclusion chromatography and fractions analyzed for the presence of BAFF by western blot using anti-human or anti-mouse BAFF antibodies, and for the activity of BAFF at the indicated dilutions of fractions using a hBAFFR:Fas reporter cell line that undergoes Fas-mediated killing upon stimulation of the chimeric receptor by BAFF. **a** Analysis of human BAFF WT and mutants. White arrows points at fractions with peak activity of BAFF 3-mer and 60-mer, as indicated. **b** Analysis of mouse BAFF WT and mutants. The experiment was performed twice for human BAFF and thrice for mouse BAFF

pair of antibodies used to monitor endogenous BAFF recognized WT mBAFF slightly better than E247K mBAFF (Supplementary Fig. 4). This indicates that circulating levels of mBAFF E247K may be equal or slightly higher than in WT, although not overwhelmingly higher.

Heterozygous (Baff[wt/E247K]) mice had a phenotype intermediate between WT and knock-in (Fig. 3c–f, Supplementary Table 1). A similar intermediate phenotype was observed in BAFF[+/−] mice (Fig. 3g, h, Supplementary Table 2), suggesting that gene dosage, and not a dominant negative action of BAFF E247K, is the most likely cause of these phenotypes.

**A model for two distinct roles of the flap of BAFF.** Knowing that (i) two BAFF 3-mers or more can interact via flap–flap interactions[19], (ii) binding to receptors does not require a functional flap (Fig. 1 and Supplementary Fig. 1), (iii) the flap is required for the activity of BAFF (Figs. 2 and 3), and (iv) recombinant human BAFF is stable either as 3-mer or 60-mer, but not as smaller intermediates such as 6-mer (Fig. 2a)[19,20,22], we propose that one BAFF 3-mer binds to three BAFFR on a B cell to form an initial complex with no or little signaling ability until two or more complexes are assembled via flap-mediated BAFF–BAFF interactions of BAFF 3-mers contained in these complexes (Fig. 4a). These flap–flap contacts require hGlu223/mGlu247, but not hHis218/mHis242.

Under certain conditions, BAFF–BAFF interactions may be stabilized into even more active 60-mer by additional contacts involving H218, at least for human BAFF (Fig. 4a)[20]. Residues participating to the core flap–flap interaction that is essential for BAFF activity, like Glu223, are conserved in BAFF of mammals, birds, reptiles, fishes, and sharks, while H218 that is required for 60-mer formation but is not essential for activity is usually not conserved in batrachians and fishes, and may not be relevant in

mice and rats (Figs. 2b and 4b). In contrast, the BAFF-like protein of sea lamprey has no flap (Fig. 4b)[27].

If the flap really serves the role of connecting BAFF 3-mers in order to activate signaling, as proposed in the model (Fig. 4a), cross-linking antibodies or other strategies to multimerize BAFF should compensate for a deficient flap.

**Cross-linking of BAFF 3-mer rescues flap defects in vitro.** The activity of size-fractionated FLAG-tagged BAFF 3-mers increased ~100-fold for human BAFF and 10-fold for mouse BAFF (WT or histidine mutants) upon anti-FLAG antibody-mediated cross-linking. The effect was even more pronounced with the glutamic acid mutants (> 1000-fold for human and > 100-fold for mouse). Upon cross-linking, all BAFF gained similarly high activities, although the glutamic acid mutant remained about 3-fold less active than the others (Fig. 4c, d). This could be explained if the cross-linking antibodies cannot arrange BAFF 3-mers in the signaling complex as precisely as via flap–flap interactions. The anti-mouse BAFF antibody 5A8 (and to a lesser extent Sandy-5) was as efficient as anti-FLAG to activate mBAFF E247K assayed on reporter cells (Fig. 4e), or on purified primary mouse B splenocytes where 5A8 fully corrected the marked signaling defect of mBAFF E247K (Fig. 4f). These results reinforce the conclusion that mBAFF E247K can bind receptors but cannot signal, and additionally demonstrated that a flap defect can be overcome by antibody-mediated cross-linking, in support of the model presented in Fig. 4a.

**Anti-BAFF antibodies rescue flap defects in knock-in mice.** Knock-in mice express flap-deficient BAFF that is competent for receptor binding but impaired in signaling (Fig. 3). When these mice were treated for 6 weeks with BAFF cross-linking antibodies

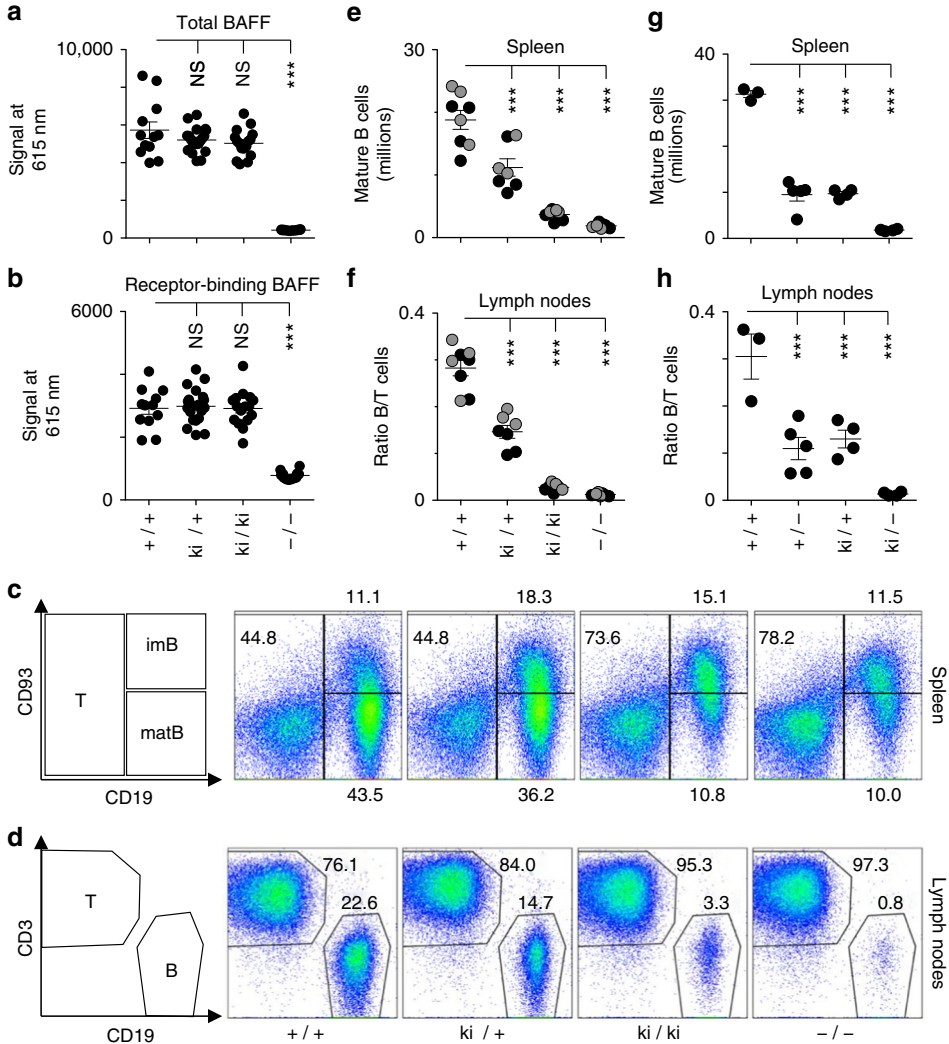

**Fig. 3** Reduced numbers of mature B cells in BAFF E247K knock-in mice despite normal circulating level of BAFF competent for receptor binding. **a** Relative levels of circulating BAFF were measured in a homogeneous ELISA-like assay (AlphaLisa) using a pair of anti-mBAFF antibodies (5A8 and Sandy-2) in the serum of $Baff^{wt/wt}$ (+/+) (n = 12), $Baff^{wt/E247K}$ (ki/+) (n = 20), and $Baff^{E247K/E247K}$ (ki/ki) (n = 16) littermates and in the serum of $Baff^{-/-}$ (−/−) (n = 12) mice. Mean ± SEM. One-way Anova followed by Bonferroni comparing +/+ to other genotypes. NS = not significant: $p > 0.05$; ***$p < 0.001$. The experiment was performed twice. **b** Same as panel **a**, except that BAFF was measured with one soluble receptor (TACI-Fc) and one anti-mouse BAFF antibody (Sandy-2). The experiment was performed once. **c** Flow cytometry analysis of lymphocytes with T cell (CD19−, CD93−), mature B cell (matB) (CD19+, CD93low), and immature B cell (imB) (CD19+, CD93+) phenotypes in spleens of mice of the indicated genotypes. Percentages of cells in each quadrant are indicated. **d** Same as panel **c**, but for T (CD3+) and B (CD19+) lymphocytes in lymph nodes. **e** Number of B cells in spleens of 7 mice per group (8 mice for WT) of the indicated genotypes from two pooled experiments differentiated by black and gray circles. Mean ± SEM. One-way Anova followed by Bonferroni comparing +/+ to other genotypes. ***$p < 0.001$. **f** Same as panel **e**, but showing B to T cell ratio in lymph nodes. **g** Number of mature B cells in spleens of mice (4 mice per group, 3 for WT, 5 for +/−) of the indicated genotypes, all littermates. Mean ± SEM. One-way Anova followed by Bonferroni comparing +/+ to other genotypes. ***$p < 0.001$. The experiment was performed once in this format (and twice comparing +/+, +/−, and −/−). **h** Same as panel **g**, but showing B to T cell ratio in lymph nodes. See also Supplementary Figs. 2, 3, and 4, and Supplementary Tables 1 and 2

(5A8 or Sandy-5), mature splenic B cells were significantly ($p < 0.05$ for Sandy-5 and $p < 0.01$ for 5A8, by one-way Anova) increased compared to the control group or to BAFF-deficient mice (Fig. 5a, b, d, Supplementary Table 3). B cells also increased in lymph nodes where the B to T cell ratio reached half or two thirds of WT levels in mice treated with Sandy-5 or 5A8, respectively (Fig. 5a, c, d).

These results reveal that knock-in mice contain a latent, inactive form of BAFF that can be reactivated in vivo by cross-linking antibodies. The repopulation of the B cell compartment proves that endogenous mBAFF E247K protein in knock-in mice can bind and activate BAFFR in vivo under adequate conditions, as seen in vitro

with the recombinant protein. This also reinforces the conclusion that the flap of BAFF has a function that is independent of, but as crucial as receptor binding for the activity of BAFF. This experiment however does not allow addressing the exact mechanism of action of activating antibodies in vivo, which is likely to depend both on activation of BAFF activity by cross-linking and on the increased half-life of mBAFF bound to antibodies.

**Belimumab and atacicept differentially inhibit BAFF 60-mer.** The anti-BAFF antibody, belimumab, is reported to inhibit soluble BAFF only[28]. We wondered whether belimumab could

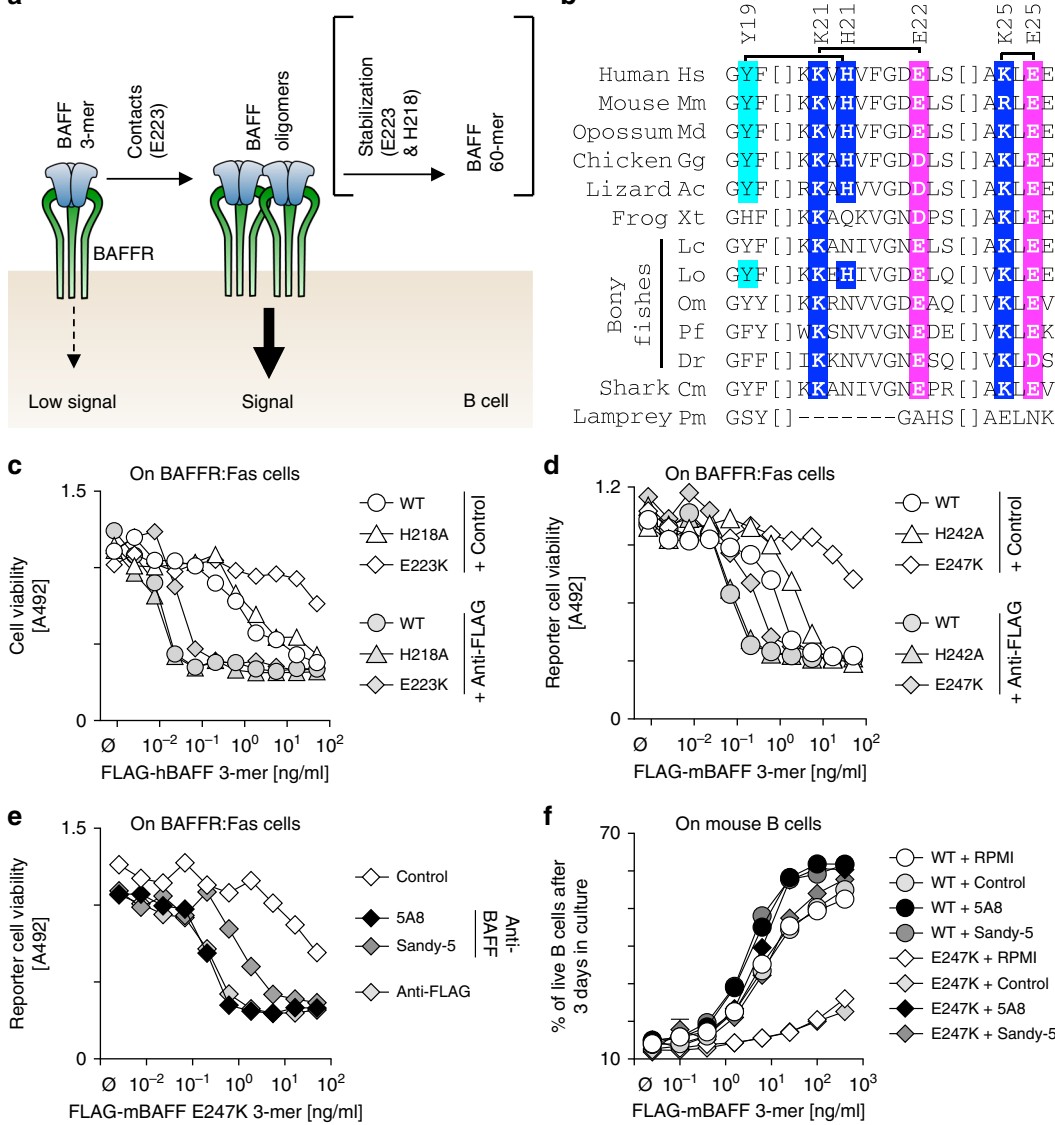

**Fig. 4** Increase and rescue of BAFF activity by cross-linking. **a** Model of the role of the flap of BAFF for activation of BAFFR signaling: flap–flap interactions for which residue E223 is important mediate interactions between receptor-bound BAFF 3-mers. This promotes BAFFR signaling. Flap–flap interactions for which residues E223 and H218 are both required can further lead to assembly of BAFF 60-mer. **b** Sequence alignment of BAFF from different vertebrates in regions relevant for 3-mer to 3-mer interactions. Residue numbering is for human BAFF. Brackets link residues with symmetric interactions (e.g., K216 of one BAFF 3-mer interacts with E223 of another BAFF 3-mer and vice-versa, see also Fig. 1a), where acidic residues are shown in magenta, basic residues in blue, and the tyrosine that pairs with His218 in turquoise. Hs: *Homo sapiens*; Mm: *Mus musculus*; Md: *Monodelphis domestica*; Gg: *Gallus gallus*; Ac: *Anolis carolinensis*; Xt: *Xenopus tropicalis*; Lc: *Latimeria chalumnae*; Lo: *Lepisosteus oculatus*; Om: *Oncorhynchus mykiss*; Pf: *Poecilia formosa*; Dr: *Danio rerio*; Cm: *Callorhinchus milii*; Pm: *Petromyzon marinus*. **c** Cell viability measured in BAFFR:Fas reporter cells exposed to the indicated concentrations of FLAG-hBAFF WT or mutants in the presence of a fixed concentration of a control antibody (EctoD1) or of an anti-FLAG cross-linking antibody. The experiment was performed twice. **d** Same as panel **c**, but for FLAG-mBAFF. The experiment was performed twice. **e** Same as panel **d**, but for FLAG-mBAFF E247K in the presence of a constant concentration of a control antibody (EctoD1), of a cross-linking antibody (anti-FLAG), or of cross-linking anti-mBAFF monoclonal antibodies (5A8 and Sandy-5). The experiment was performed twice. **f** Cell viability of primary splenic mouse B cells cultured for 3 days in the presence of titrated amounts of FLAG-mBAFF WT or E247K, alone or in the presence of fixed amounts of control (EctoD1) or cross-linking (5A8, Sandy-5) antibodies. Each point is the mean ± SEM of technical triplicates. Experiment performed once in this format (and once with less titration points)

target the flap of BAFF to prevent its signaling function. However, both belimumab and atacicept (TACI-Fc) efficiently inhibited Fc-BAFF, an intrinsically cross-linked (and therefore flap-independent) BAFF, indicating that belimumab is unlikely to target only the flap of BAFF (Fig. 6a, b). However, belimumab did not inhibit naturally cleaved human BAFF, even when belimumab was used at concentrations higher than those of atacicept (Fig. 6c, d).

As naturally processed human BAFF can form 60-mers (Fig. 2a)[22], we tested belimumab activity on purified BAFF 60-

mer, which belimumab also failed to inhibit (Fig. 6e, f), in line with another report[29]. These experiments strongly suggest that belimumab, unlike atacicept, does not (immediately) inhibit BAFF 60-mer.

**Belimumab inhibits BAFF 60-mer after dissociation to 3-mer.** As BAFF 60-mers are in equilibrium with BAFF 3-mers[19,22], belimumab should be able to inhibit BAFF 60-mer indirectly by

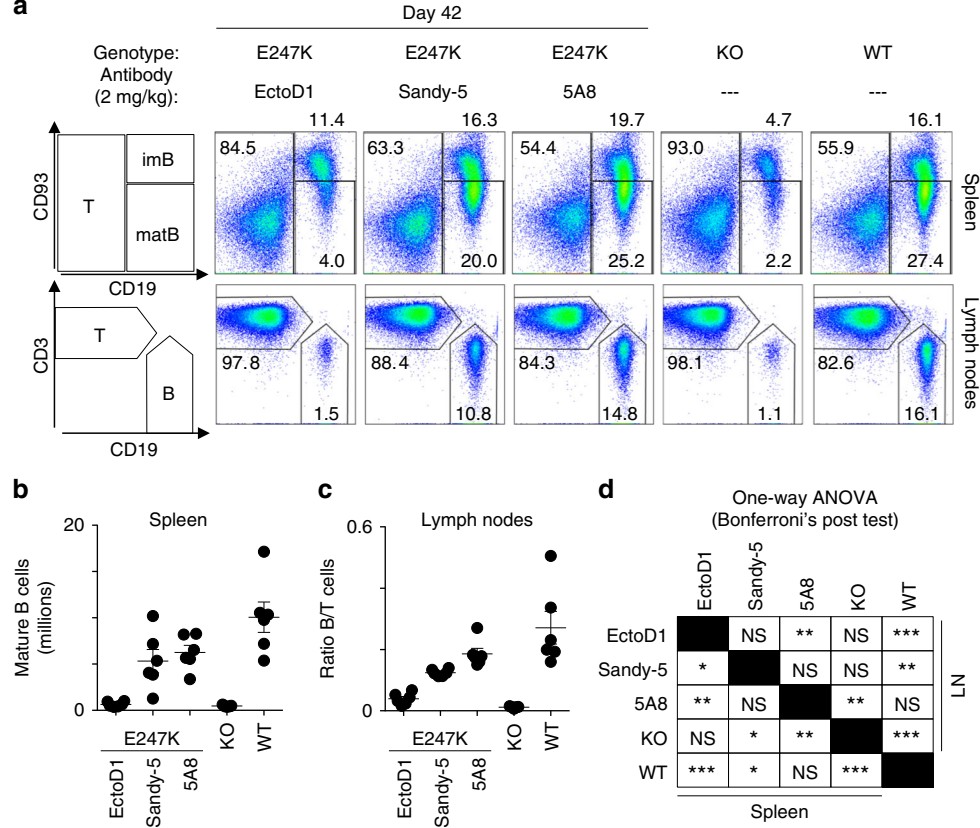

**Fig. 5** Restoration of mature B cells in BAFF E247K knock-in mice treated with cross-linking anti-mBAFF antibodies. Cohorts of BAFF E247K knock-in mice received control (EctoD1) or BAFF cross-linking antibodies (Sandy-5 or 5A8) weekly for 6 weeks, then spleen and lymph nodes were analyzed for the presence of mature B cells. Groups of untreated wild type (WT) and BAFF knock-out (KO) were analyzed in parallel. **a** Flow cytometry analysis of spleen (top dot plots) and lymph nodes (bottom dot plots). Cell populations analyzed are shown on the left. imB: immature B cells; matB: mature B cells. **b** Same as panel **a**, but showing number of B cells in spleens of 6 mice per group (5 mice for KO). Mean ± SEM. For statistics, see panel **d**. **c** Same as panel **b**, but showing B to T cell ratio in lymph nodes. For statistics, see panel **d**. Experiment of panels **a**–**c** was performed once in this format (and once with 5A8 in two ki mice, looking at kinetics of successful B cell restoration in blood). **d** Results of one-way Anova followed by Bonferroni multiple comparison test for mature B cells in the spleen (lower left half) and B/T cell ratio in lymph nodes (upper right half). NS = not significant: $p > 0.05$; *$p < 0.05$; **$p < 0.01$; ***$p < 0.001$. Panels **b**–**d**: results of statistical test should be considered with caution because although most measures were distributed normally, the assumption of equal variance between groups was not met. See also Supplementary Table 3

capturing BAFF 3-mers released upon dissociation. To test this hypothesis, BAFF 60-mer was incubated for 3 days at 37 °C with or without belimumab or atacicept. BAFF 60-mer alone retained full activity under these conditions (Fig. 6g) and atacicept inhibited BAFF 60-mer both directly and after co-incubation (Fig. 6h). Belimumab did not significantly inhibit BAFF 60-mer when used directly, but could do so, after a co-incubation of 3 days, although still less efficiently than atacicept (Fig. 6i). In another experiment, belimumab was incubated with an excess of BAFF 60-mer, with the expectation that binding to BAFF 60-mer or to BAFF 3-mer would differently shift the size of belimumab. In this experiment, the elution position of belimumab was indeed shifted, but remained smaller than that of BAFF 60-mer still present in the mixture (Supplementary Fig. 5). Taken together these results indicate that belimumab cannot inhibit BAFF 60-mer directly, but can eventually do so when BAFF 60-mer dissociates into 3-mers.

**3D elucidation of the mechanism of action of belimumab**. The antigen-binding fragment (Fab) of belimumab was produced by digestion with papain, while BAFF 3-mer was expressed in prokaryotic cells. To prevent association of BAFF 3-mer into 60-mer, mutation H218A was introduced into the flap region[22], which did not grossly affect the affinity of belimumab for BAFF (Supplementary Fig. 6). BAFF 3-mer or the Fab taken alone had roughly equal native sizes (Fig. 7a–c), but formed a larger, well-defined complex when BAFF was mixed with an excess of belimumab Fab (Fig. 7d). Some free Fab was detected when the complex was re-analyzed by size-exclusion chromatography (Fig. 7e), suggesting that the Fab was not very tightly bound to BAFF.

As expected, the high-molecular-weight complex contained both BAFF and the Fab when analyzed by reducing or non-reducing SDS-PAGE (Fig. 7f). The BAFF–Fab complex was analyzed by both electron microscopy and crystallography (Supplementary Fig. 7, Supplementary Table 4). Negative stain electron microscopy images often revealed structures with three arms on which the crystal structure could readily be super-imposed (Fig. 7g, h). The complex sometimes also appeared as L-shaped, which could be interpreted either as a complex with 3 Fab, one of which would stand perpendicular to the image plane, or as a BAFF 3-mer bound to only two Fab (Fig. 7i, j). Incomplete complexes with a single Fab bound to a BAFF 3-mer could also be seen (Fig. 7k). This less abundant form was likely excluded from crystals where only BAFF bound to three Fabs could be observed. Interestingly, the relative instability of the BAFF–Fab complex

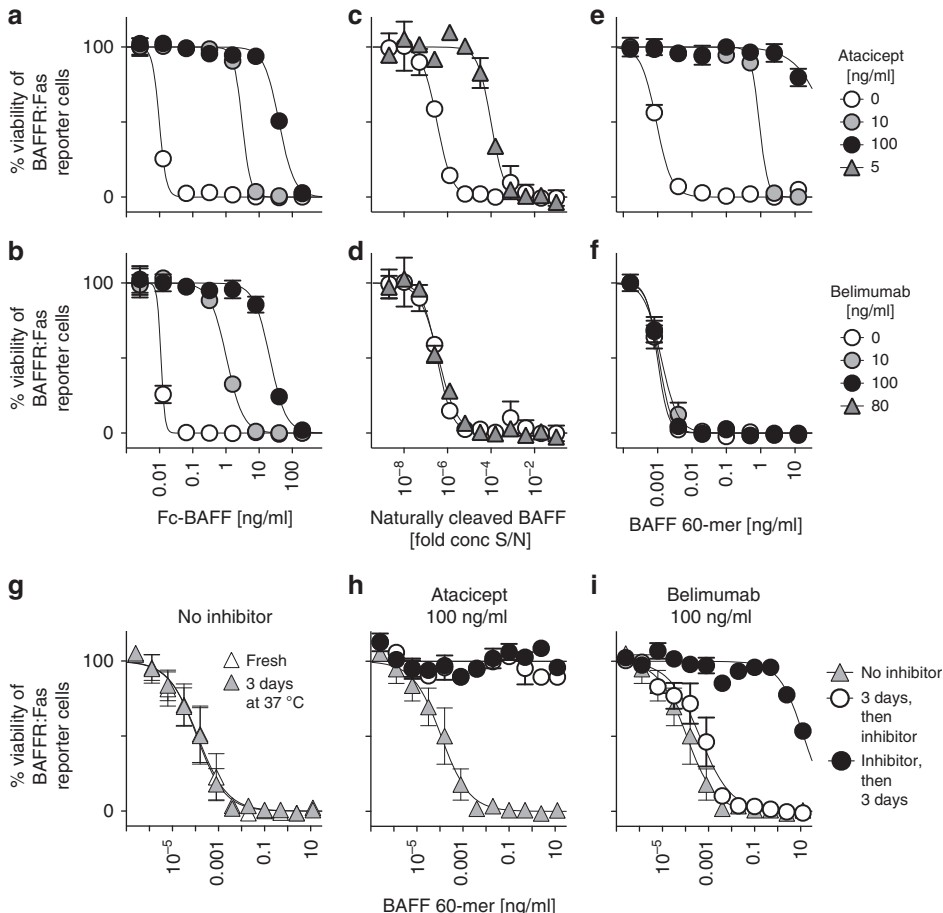

**Fig. 6** Belimumab differs from atacicept in that it does not inhibit BAFF 60-mer promptly. BAFF-responsive BAFFR:Fas reporter cells were treated overnight with titrated amounts of the indicated forms of BAFF, in the presence or absence of the inhibitors atacicept or belimumab at fixed concentrations. Cell viability was then monitored. **a, b** Inhibition of Fc-BAFF by atacicept (**a**) or belimumab (**b**). The experiment was performed 3 times. **c, d** Naturally processed hBAFF in supernatants of transfected 293 T cells exposed to atacicept at 0 or 5 ng/ml (**c**) or belimumab at 80 ng/ml (**d**). The experiment was performed twice. **e, f** Inhibition of Fc-BAFF by atacicept (**e**) or belimumab (**f**). The experiment was performed 3 times. **g** BAFF 60-mer was titrated in medium, then incubated in medium for 3 days at 37 °C, then added to reporter cells. Alternatively, BAFF 60-mer was titrated in medium just before the assay (fresh). **h, i** BAFF 60-mer was titrated in medium without or with atacicept (**h**) or belimumab (**i**) for 3 days at 37 °C. Samples without inhibitors received inhibitor or not after 3 days of incubation. Samples were then added to reporter cells. The experiment was performed twice. Panels **a**, **b**, **e**, and **f** show the mean ± SEM of 5 replicates, panels **c** and **d** of duplicate, and panels **g**, **h** and **i** of triplicate measures. The condition without inhibitor is the same in panels **a** and **b**, in panels **c** and **d**, in panels **e** and **f**, and in panels **g**, **h**, and **i**. See also Supplementary Fig. 5

was suggested by the observation that the purified complex was almost as active as BAFF alone on reporter cells, which could be explained either by the existence of a small proportion of free BAFF, or by competition of high-affinity BAFFR with Fab for a common binding site on BAFF (Fig. 7l).

A closer look at the structure of the BAFF–belimumab Fab complex showed that both heavy and light chains of belimumab contacted BAFF over its entire height, forming two distinct interfaces with the receptor-binding site (664 Å²) and the flap region (333 Å²), for a total average surface area of 995 Å² (963–1014 Å² for the six copies of the asymmetric unit) (Fig. 8a). In contrast, in a model of TACI bound to BAFF, the receptor bound to a much smaller region of BAFF (Fig. 8b) that was clearly separated from the flap–flap interaction region mediating BAFF 60-mer formation (Fig. 8c).

The BAFF–Fab complex is the first structure in which BAFF 3-mers make no contact with adjacent BAFF 3-mers via flap–flap interactions. Nevertheless, the conformation of the flap loop was virtually identical to that of a flap engaged in 3-mer to 3-mer contacts, indicating that its conformation is constitutive and not induced by binding (Fig. 8d). In addition, mutation H218A did

not grossly alter conformation of the flap (Fig. 8d). Belimumab made important contacts with a small cavity of BAFF that normally accommodates the DXL loop motif present in all natural receptors of BAFF and that is essential for receptor–ligand interactions[30]. In fact, one of the variable loops of the heavy chain of belimumab contained a DXL (DLL) motif that perfectly mimicked the receptor-binding site, while the rest of the antibody had no structural similarity with TACI or other BAFF receptors (Fig. 8e, Supplementary Fig. 8). Finally, the light chain of belimumab contacted a region of BAFF not involved in interactions with the natural receptors. Fitting the structure of the Fab of belimumab on BAFF 60-mer resulted in a clash area of about 700 Å² with BAFF (Fig. 8f) that would preclude binding of belimumab to BAFF 60-mer. Taken together, these data show that belimumab hijacks the natural receptor-binding site of BAFF via a DXL motif in its heavy chain, but that the bulky nature of the Fab and the additional contacts made by the light chain with BAFF prevent interactions with BAFF 60-mer.

**Atacicept binds to the surface of intact BAFF 60-mers.** BAFF 60-mer migrated with a high molecular weight by size-exclusion

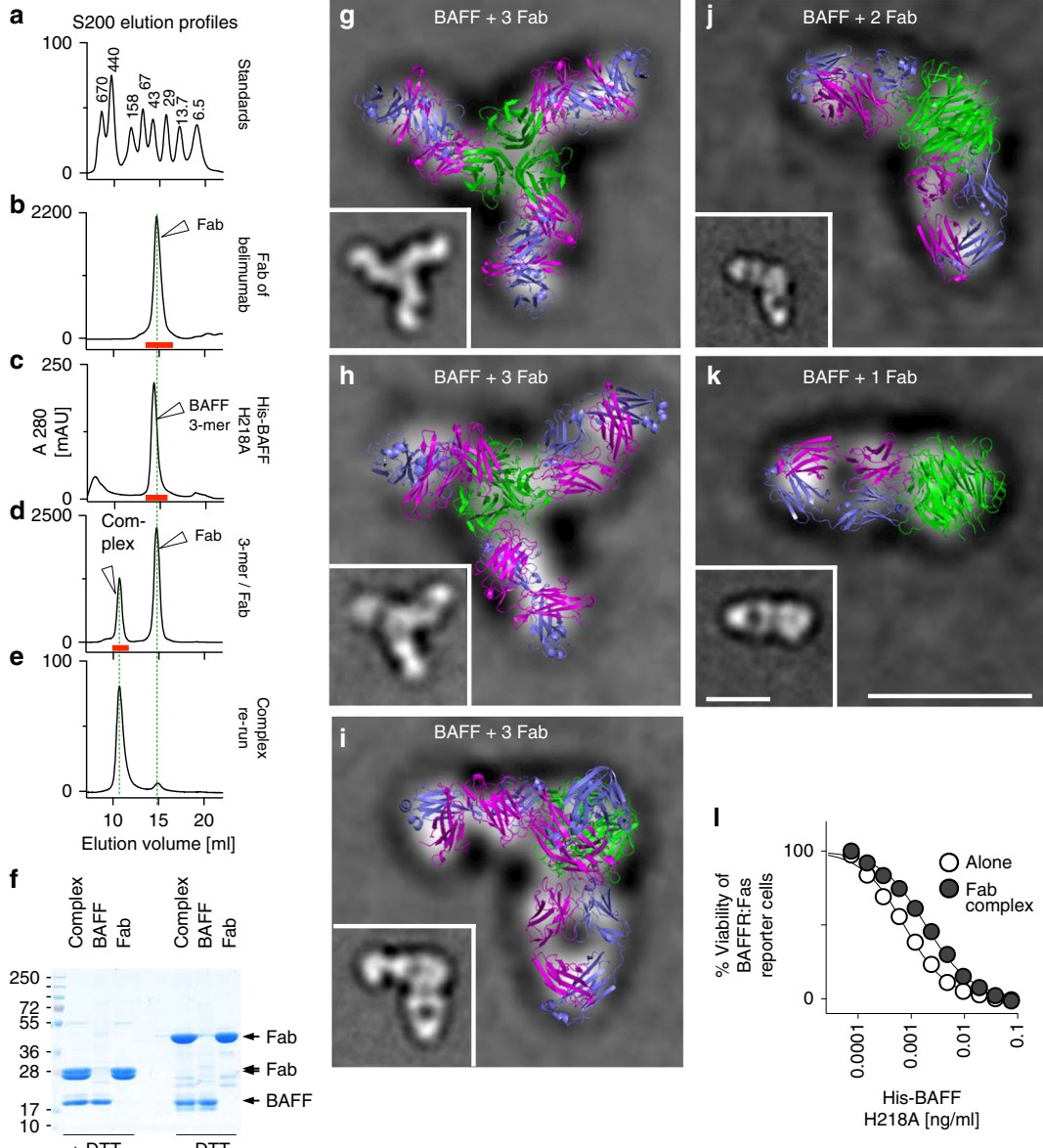

**Fig. 7** Structure of the BAFF–belimumab Fab complex. The Fab of belimumab was generated by papain digestion, and a 3-mer of BAFF was obtained by expression of BAFF H218A in bacteria. The complex formed by the association of BAFF 3-mer with belimumab Fab was isolated by size-exclusion chromatography and characterized by electron microscopy and by crystallography. The activity of BAFF 3-mer alone or in complex with belimumab Fab was tested on BAFFR:Fas reporter cells. **a** Elution profile of molecular weight standards by size-exclusion chromatography. **b** Elution profile of belimumab Fab. The experiment was performed twice. **c** Elution profile of His-BAFF H218A. The experiment was performed twice. **d** Elution profile of BAFF 3-mer mixed with an excess of belimumab Fab. The experiment was performed twice. For panels **b**, **c**, and **d**, collected fractions are indicated by the underlying red line. **e** Elution profile of the isolated BAFF–belimumab Fab complex. The experiment was performed twice. **f** 5 µg of BAFF 3-mer, 15 µg of belimumab Fab, and 20 µg of the BAFF–Fab complex were analyzed by reducing (+DTT) or non-reducing (−DTT) SDS-PAGE and Coomassie blue staining. The migration positions of BAFF and of the Fab chains are indicated on the right. The experiment was performed twice. **g**–**k** Superposition of typical electron microscopy negative stain images with the crystal structure of the BAFF–belimumab Fab complex. Note that in panels **j** and **k**, respectively one and two Fab were removed from the crystal structure. The experiment was performed once. Scale bars: 100 Å. **l** Biological activities of BAFF 3-mer and of BAFF–belimumab Fab complexes were measured on reporter cells. The experiment was performed three times. See also Supplementary Table 4

chromatography (Fig. 9a, b). When BAFF 60-mer was mixed with an excess of ataticept on ice, a heavy precipitate formed, and the supernatant only contained excess soluble ataticept with no trace of complex (Fig. 9c). However, when resuspended in buffer, the precipitated complex reversibly dissolved at room temperature and re-precipitated on ice (Fig. 9d). The ataticept–BAFF 60-mer complex could therefore be isolated by size-exclusion chromatography at room temperature (Fig. 9e, f). Analysis of BAFF 60-mer by electron microscopy revealed a well-

organized, capsid-like structure of about 220 Å diameter (Fig. 9g, pictures on the left)[19]. In the presence of ataticept, these particles enlarged to a diameter of about 340 Å, indicating that they had been coated with ataticept without affecting assembly of the underlying BAFF 60-mer (Fig. 9g, pictures on the right). We conclude from these studies that although ataticept and belimumab share common features for BAFF binding (the DXL motif), they differ by their binding ability to un-dissociated BAFF 60-mer.

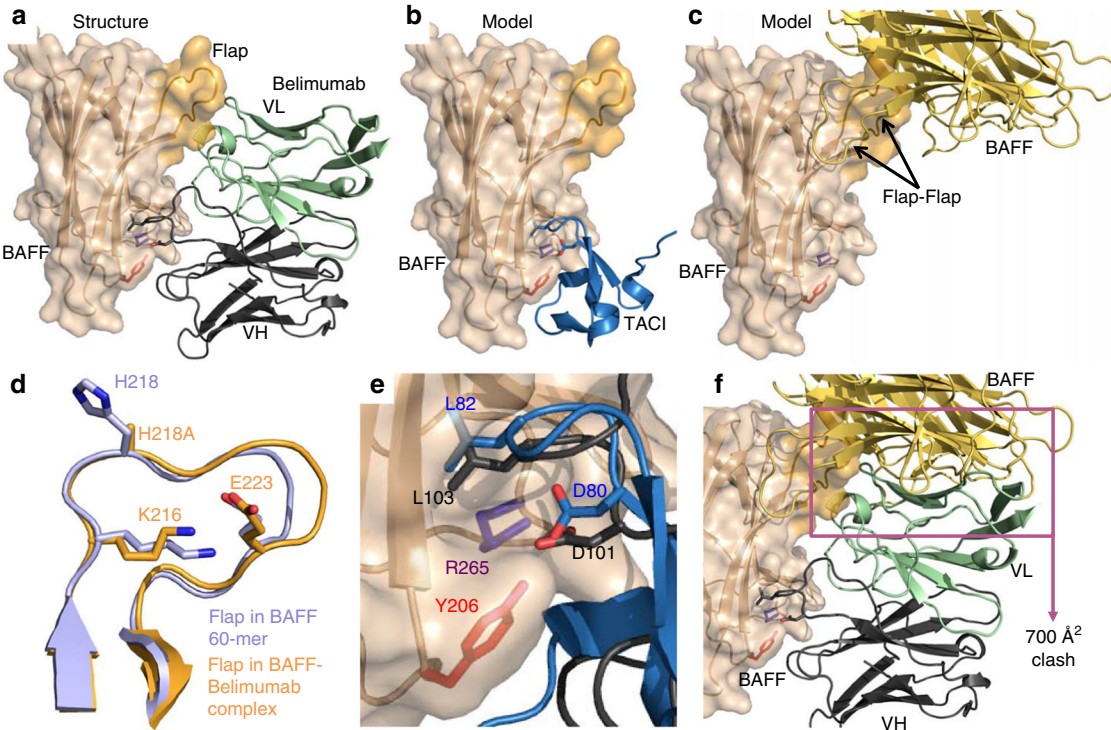

**Fig. 8** Prominent features of the BAFF 3-mer–belimumab Fab complex. The structure of BAFF H218A (3-mer) in complex with the Fab of belimumab was solved. The experiment was performed once. A model of the cysteine-rich domain 2 of TACI bound to BAFF was obtained by superposing the structure of one BAFF monomer to the structure of APRIL in the APRIL–TACI complex (PBD-ID 1XU1). **a** Structure of one BAFF monomer (pale yellow) to one belimumab Fab (heavy chain in black, light chain in green). The flap of BAFF is shown in darker yellow. **b** Model of TACI bound to BAFF. **c** Model showing the position of the adjacent BAFF monomer if a flap–flap interaction would have occurred. The model was built by superimposing two relevant BAFF monomers from the BAFF 60-mer structure (PDB-ID 1OQE) to one BAFF monomer of the BAFF–belimumab Fab complex. **d** Superimposition of the flap of a BAFF from the BAFF 60-mer structure (pale purple) (PDB-ID 1OQE) and a flap from the BAFF H218A–belimumab Fab complex (orange). Residues K216 and E223 that make important contacts with E223 and K216 of an adjacent flap in the BAFF 60-mer structure are shown. **e** Detail of the receptor-binding pocket of BAFF, showing interaction with the DXL motif of belimumab (D101, L103). The DXL motif of TACI (D80, L82) (PDB-ID 1XU1) is superimposed. Y206 and R265 of BAFF are also shown. **f** The structure of the BAFF–Fab complex of panel **a** was merged to that of the flap-mediated BAFF–BAFF interaction of panel **c**. The large clash area between both structures suggests that belimumab cannot access its binding site in BAFF 60-mer

**Potential impact of distinct anti-BAFF drugs specificities**. In the present study, the structural basis for the distinct inhibitory mechanisms of BAFF 60-mer by belimumab and atacicept was characterized. This is not the only difference between belimumab and atacicept regarding their substrate specificity: atacicept, but not belimumab, also inhibits APRIL, and heteromers of BAFF and APRIL[31]. Whether these differences positively or negatively impact the therapeutic action of these drugs is unknown. This will depend on whether drug targets are expressed or not in the pathological context, and also on how detrimental pathogenic cells (e.g., autoreactive or pro-inflammatory cells) and useful protective cells (e.g., normal plasma cells or regulatory cells) overlap with their requirements for potentially distinct sets of survival factors and for the set of receptors (BAFFR, TACI, and BCMA) used to respond to them. For example, if a subset of autoreactive plasma cells are pathogenic in human SLE, and these cells require either BAFF or APRIL, like normal mouse plasma cells[32], atacicept could be superior to belimumab. But if this same pathogenic cell subset would turn out to require BAFF only, atacicept could induce an unnecessary immunodeficiency. With regard to BAFF 60-mer, about which almost nothing is known in humans, much work is still needed to understand whether and where this complex occurs, and for which physiological or pathological functions. Also, the different mechanisms of BAFF 60-mer inhibition characterized in this study could be irrelevant in vivo if BAFF 60-mer dissociates into 3-mers before it can act on target cells, or crucial if newly synthetized BAFF 60-mer can reach its target before dissociation.

**Modulation of TNF family ligand activity by oligomerization**. It has long been recognized that membrane-bound and soluble TNF differentially activate TNFRSF1A (TNFR1) and TNFRSF1B (TNFR2) and fulfill different functions in inflammation and immunity to mycobacteria[33–35]. Similarly, membrane-bound and soluble TNF ligand superfamily member 6 (FasL) have different capacities to induce either apoptosis or non-deadly functions such as cell migration (reviewed in ref. [36]). Membrane-bound ligands can only deliver signals in a cell-to-cell contact but can activate receptor functions more efficiently than soluble ligands. In contrast, soluble ligands can act systemically but require receptors responsive to such soluble ligands. The plasma membrane is believed to cluster membrane-bound ligands, a function that can be mimicked when soluble ligands are either cross-linked or engineered to contain two trimers within a single molecule[37]. Ectodysplasin-A (EDA), the only TNF family ligand containing a collagen domain, must be processed to a soluble form to fulfill its functions in the embryonic development of skin-derived appendages[38]. In this case, it is the collagen domain that mediates an intrinsic oligomerization to make soluble EDA efficient activators of their oligomerization-sensitive receptor[39]. BAFF is another ligand with an intrinsic propensity to oligomerize, but this time as a 60-mer. BAFF 60-mer, or membrane-bound BAFF, might be

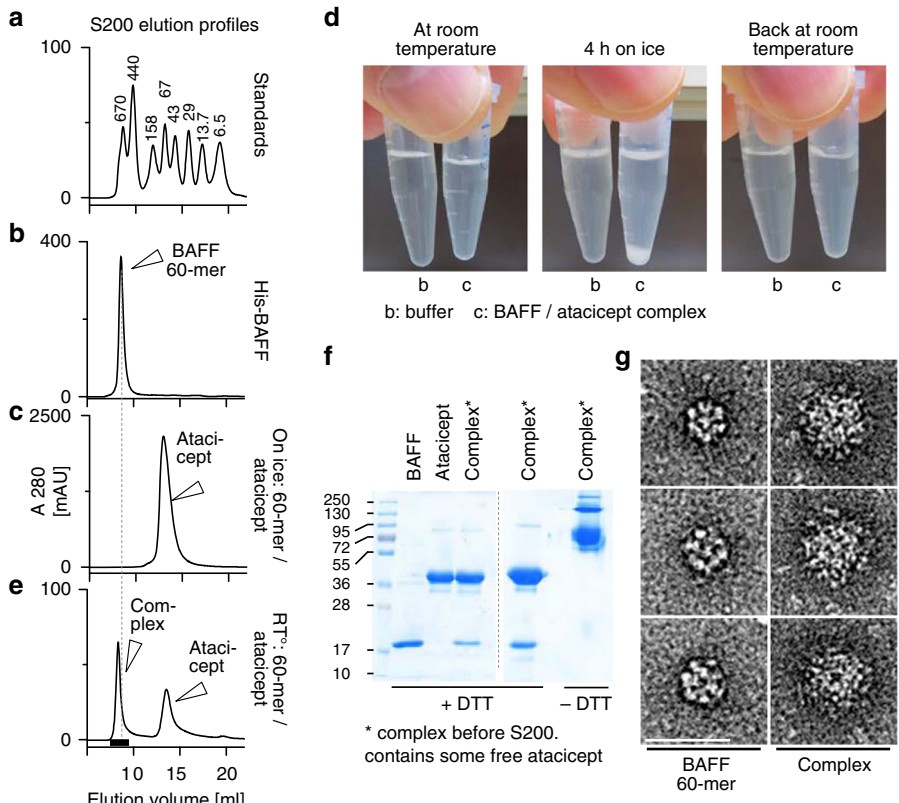

**Fig. 9** Binding of atacicept to BAFF 60-mer does not induce BAFF dissociation. BAFF 60-mer was mixed with an excess of atacicept to form a complex that was soluble at room temperature but that reversibly precipitated on ice. The atacicept–BAFF 60-mer complex was isolated by size-exclusion chromatography and analyzed by electron microscopy. **a** Elution profile of molecular weight standards by size-exclusion chromatography (same as those shown in Fig. 7a). **b** Elution profile of purified BAFF 60-mer. The experiment was performed twice. **c** Elution profile of atacicept in the supernatant of the BAFF 60-mer–atacicept precipitate. Note that no high-molecular-weight complex can be detected. This analysis was performed once. **d** The mixture of BAFF 60-mer and atacicept is soluble at room temperature, but precipitates and spontaneously sediments on ice. The precipitate re-dissolves if samples are warmed back at room temperature. The experiment was performed three times. **e** Elution profile of the BAFF 60-mer–atacicept complex run at room temperature. This analysis was performed once. **f** SDS-PAGE and Coomassie blue analysis of BAFF 60-mer (5 µg), atacicept (15 µg), and the BAFF 60-mer–atacicept complex (re-solubilized precipitate, before size-exclusion chromatography; ~17 µg). The complex was also analyzed under reducing (+DTT) and non-reducing (−DTT) conditions. This analysis was performed twice. **g** Three representative negative stained electron microscopy pictures of BAFF 60-mer alone (left column) or of BAFF 60-mer–atacicept complexes (right column). This analysis was performed once. Scale bar: 400 Å

required to activate TACI and possibly BCMA[20]. In contrast, BAFFR can also be stimulated by BAFF 3-mer and, in that respect, resembles TNFR1 activated by soluble TNF. We now suggest that the flap of BAFF clusters individual complexes of BAFF 3-mer/BAFFR to form multimers endowed with the capacity to initiate relevant B cell survival signals in vitro and in vivo. Clustering of individual BAFF-BAFFR signaling units probably represents the primary and *sine qua non* function of the flap of BAFF, while the capacity of the flap to assemble BAFF 60-mer could be a secondary improvement of this oligomerizing function.

In summary, binding and activation of BAFFR by BAFF are distinct steps. With regard to BAFF-directed therapies, the flap of BAFF should be considered as a target potentially as good as the receptor-binding site for BAFF inhibition. Conversely, stimulating BAFF activity by promoting oligomerization could prove beneficial in certain immune deficiencies. Finally, the flap of BAFF not only controls BAFF signaling (via 3-mer to 3-mer interactions) and BAFF potency (via 60-mer formation), but can also interfere with the binding and action of anti-BAFF drugs.

## Methods

**Animals.** $Baff^{-/-}$ mice were as described[12,20]. C57BL/6OlaHsd mice were purchased from Envigo (Horst, Netherlands, stock# 057) and housed in a specific

pathogen-free animal facility. BAFF E247K knock-in mice were generated according to standard procedures. Briefly, embryonic stem cells (129 SvEv) were electroporated with a construct containing the E247K mutation (GAG CTG → AAG CTT) in exon 6 of *Baff* and a floxed NeoR cassette inserted in introns 6–7 and screened by PCR and Southern blot for homologous recombination events (Supplementary Fig. 2). Recombined embryonic stem cells were injected into C57BL/6 blastocysts followed by transfer into the uterus of pseudo-pregnant recipients. Progeny of chimeric mice that transmitted the mutation was backcrossed for 4 generations onto the C57BL/6 background, then bred to B6.C-Tg(CMV-cre)1Cgn/J mice (sn#006054, The Jackson Laboratory) to remove the NeoR cassette. Heterozygous mice were bred to obtain cohorts of $Baff^{wt/wt}$, $Baff^{wt/E247K}$, and $Baff^{E247K/E247K}$ littermates, or bred to $Baff^{-/-}$ to obtain cohorts of $Baff^{wt/wt}$, $Baff^{wt/-}$, $Baff^{wt/E247K}$, and $Baff^{E247K/-}$ littermates. Experiments were performed according to guidelines and under the authorization of the Swiss Federal Food Safety and Veterinary Office (authorization 1370.6 to PS).

**Genotyping.** Ear biopsies were digested for 2–16 h at 55 °C in 50 µl of the DirectPCR Lysis Reagent (Peqlab, 31-102-T) plus 2 µl of Proteinase K (Roche), followed by 45 min at 85 °C. Supernatants of BAFF-E247K ki lysates were used for PCR using primers fwd 5′-ACCCTGTTCCGATGTATTCA-3′ and rev 5′-TAA-GAGAGTGCCAGGTCCC-3′, with 30 cycles of 94 °C (7 s)/58 °C (20 s)/72 °C (40 s). Supernatants of BAFF-ko lysates were used for PCR using primers shared 5′-GCAGATTGAGCAATCCATGGAAGGCCA-3′, wt specific 5′-CAAGTTGATG TCCTGACCCAAGGCACC-3′, and neo 5′-TGGCAGGGTCTTTGCAGACTC ATCCAT-3′, with 30 cycles of 94 °C (7 s)/60 °C (20 s)/72 °C (20 s). For the screening of recombined ES cells, genomic DNA was amplified with primers NeoR fwd 5′-CCTTCTATCGCCTTCTTGAC-3′ and BAFF rev 5′-GTGGAACAGATAA GGTGCCT-3′, with 3 cycles of 95 °C (30 s)/64 °C [ramp-0.5 °C/cycle] (30 s)/68 °C

(2 min), followed by 30 cycles of 95 °C (30 s)/54 °C (30 s)/68 °C (2 min), using TaKaRa LA polymerase (TaKaRa) at 5 U/ml.

**Reagents and cell lines**. The following antibodies were used: mouse IgG1 anti-FLAG M2 and biotinylated M2 (Sigma Aldrich F3165 and F9291), goat IgG 852 anti-mouse BAFF (R&D AF2106), biotinylated mouse IgG1 anti-human BAFFR huBR9.1 (Adipogen, Liestal, Switzerland, AG-20B-0016B), rat IgG1 anti-mouse BAFF 5A8[40], mouse IgG1 anti-mouse BAFF Sandy-2 and Sandy-5[41], rat IgM anti-human BAFF Buffy-2[9], and mouse IgG1 anti-EDA EctoD1[42]. Human IgG1 anti-human BAFF antibody belimumab (registered trade name Benlysta) was purchased from the Pharmacy of Lausanne University Hospital (CHUV). hBAFFR-Fc was from Adipogen (AG-40B-0027) and hTACI (aa 31–110)-hIgG1 Fc (aa 245–470, L258E, A353S, P354S) (atacicept) was provided by Merck, KGaA. HEK 293, HEK 293 T, Jurkat, and BJAB cells were obtained from late Jürg Tschopp (University of Lausanne). CHO-S cells were from Thermoscientific (A1155701). Reporter cells Jurkat-JOM2-hBAFFR:Fas clone 21[31,43,44] were as described and grown in RPMI supplemented with 10% fetal calf serum (FCS). HEK 293 T cells were grown in DMEM 10% FCS. BJAB and BJAB–TACI cells[45] were grown in RPMI supplemented with 10% FCS. Cell lines were tested for mycoplasma using MycoAlert mycoplasma detection kit (Lonza, LT07-318) and found to be negative. As the identity of CHO-S, HEK 293, HEK 293 T, and Jurkat cells does not impact on result interpretation, these cell lines were not authenticated. The identity of BJAB cells was confirmed by microsatellite sequencing (cell line typing service, Microsynth, Balgach, Switzerland).

**Immunoglobulin-depleted fetal calf serum**. A volume of 500 ml of FCS was depleted from immunoglobulins by repeated passages (usually eight times) at 4 ml/min on a 5 ml Protein A-Sepharose column (GE Healthcare), until the fraction of immunoglobulins bound to the column became negligible. Immonoglobulin-depleted serum was sterilized by filtration at 0.2 μm.

**Purification of Fc-containing recombinant protein**. Fc-containing proteins (from plasmids listed in Supplementary Table 5) were produced from stable clones of HEK 293 or CHO-S cells. Cells were grown in serum-free OptiMEM medium or in DMEM/F12 (1:1, v/v) medium supplemented with 2% of immunoglobulin-depleted FCS. Proteins in conditioned supernatants were affinity-purified on 5 ml (or 1 ml) Protein A- or Protein G-Sepharose column (GE Healthcare), eluted with 50 mM citrate-NaOH pH 2.7 and neutralized with appropriate amounts of 1 M Tris–HCl pH 9. Proteins were concentrated, and buffer exchanged for PBS using 30 kDa cutoff centrifugal devices, then sterilized by filtration at 0.2 μm, quantified by absorbance at 280 nm using theoretical molar extinction coefficients and stored at −70 °C until use[44,46].

**Purification of His-tagged BAFF**. *Escherichia coli* M15 pRep4 bacteria expressing His-tagged hBAFF H218A (3-mer) and His-tagged hBAFF (60-mer)[21] in early logarithmic phase of growth were grown overnight at 18 °C in L-Broth medium supplemented with 0.1 mM of isopropyl-thiogalactoside. Bacteria were harvested, lysed by sonication in 30 ml of 10 mM Tris–HCl pH 8, 0.5 M NaCl per liter of culture and insoluble material was removed by centrifugation (15 min, 17,000 × g). Supernatants were loaded (up to three liter equivalent of culture) on a 5 ml chelating-Sepharose column (GE Healthcare) pre-loaded with 0.5 M ZnSO₄ and equilibrated in lysis buffer. The column was washed with two volumes of 10 mM Tris–HCl pH 7.4, 150 mM NaCl (TBS), five volumes of TBS 50 mM imidazole, two volumes of TBS, and eluted with three volumes of TBS, 50 mM ethylenediaminetetraacetic acid (EDTA). Proteins were concentrated (30 kDa cutoff) and fractionated by size-exclusion chromatography on a Superdex 200 column as described under 'size-exclusion chromatography'. Fractions corresponding to BAFF 3-mer or BAFF 60-mer were filter-sterilized, quantified by absorbance at 280 nm and stored at −70 °C.

**Generation of the Fab fragment of belimumab**. Quantity of 120 mg of belimumab was mixed with 500 μl of immobilized papain beads (ThermoScientific) in 3 ml of 20 mM cysteine, 20 mM Na-phosphate pH 7, 10 mM EDTA, and digested for 4 days at 37 °C. At the end of the incubation, beads were removed and the Fab was recovered in the flowthrough of a Protein A affinity column. Fab were concentrated on 10 kDa cutoff centrifugal devices, then size-fractionated on a Superdex 200 Increase column in 20 mM Hepes pH 7.5, 130 mM NaCl.

**Relative affinity of belimumab Fab for Fc-BAFF WT and H218A**. Quantity of 300 μg of belimumab Fab in 300 μl of 0.1 M Na-borate pH 8.8 was biotinylated for 2 h at room temperature with 3 μl of sulfo-N-hydroxysuccinimide-LC-biotin (Pierce, 21335) at 30 μg/ml in DMSO. Reaction was terminated by addition of 10 μl of 1 M NH₄Cl, and buffer was exchanged to PBS using a 30 kDa cutoff centrifugal device. Fc-hBAFF (WT or H218A, 100 μl) was coated at 1 μg/ml in PBS in an ELISA plate. After blocking, 50 μl of soluble Fc-BAFF (WT or H218A) were added at 2-fold the desired final concentration, followed by the addition of 50 μl of 100 ng/ml biotinylated Fab of belimumab and immediate mixing. After washing, bound Fab was detected with horseradish peroxidase-coupled streptavidin.

**Complex between BAFF 3-mer and the Fab of belimumab**. Quantity of 8.8 mg of His-hBAFF H218A was mixed with 50 mg of the Fab fragment of belimumab in 20 mM Hepes pH 7.5, 130 mM NaCl, incubated for 16 h at 4 °C, and size-fractionated by size-exclusion chromatography in the same buffer. Fractions of complex were concentrated to 28 mg/ml.

**Generation of a complex between BAFF 60-mer and atacicept**. Quantity of 0.9 mg of His-hBAFF 60-mer at 4.6 mg/ml was mixed with 9.7 mg atacicept at 75 mg/ml in 20 mM Hepes pH 8.2, 130 mM NaCl and incubated for 1 h on ice. The precipitate was recovered by centrifugation at 4 °C, then re-dissolved at room temperature in 20 mM Hepes pH 8.2, 130 mM NaCl and size-fractionated at room temperature by size-exclusion chromatography in the same buffer. Fractions of the complex were concentrated to 0.8 mg/ml.

**Transfections**. HEK 293 T cells were transiently transfected with the poly-ethyleneimide method[47] and grown for 7 days in serum-free OptiMEM, after which time supernatants were harvested and concentrated 20 times in 30 kDa cutoff centrifugal concentration devices.

**Generation of BAFFR-deficient cell lines**. BJAB and BJAB-TACI cells deficient for BAFFR were generated by lentiviral transduction of a CRISPR/Cas9-expression vector carrying a hBAFFR gRNA (Supplementary Table 5). Annealed oligonucleotides 5′-CACCGGGCCGAGTGCTTCGACCTGC-3′ and 5′-AAACGCAGGTCGAAGCACTCGGCCC-3′ were cloned in the BsmBI restriction site of lentrcrispr v2 plasmid (Supplementary Table 5)[48]. This plasmid was co-transfected with co-vectors pCMV-VSV-g and psPAX2 (Supplementary Table 5) into 293 T cells with polyethyleneimide. The next day, cells were washed with PBS and cultured for an additional day in RPMI 10% FCS. Cell supernatants filtered at 0.45 μm were supplemented with 8 μg/ml polybrene (Sigma, H9268) and 3 ml were added to 2 × 10⁶ pelleted target cells that were subsequently cultured overnight in a 12-well plate, then expanded for 2 days in a 6-well plate. Cells were then submitted to two rounds of selection in RPMI 10% FCS, 1 μg/ml puromycin (EnzoLi-feSciences). Surviving cells were cloned in RPMI 10% FCS. BAFFR-deficient clones were identified by flow cytometry after staining with biotinylated huBR9.1 at 2 μg/ml, followed by PE-coupled streptavidin at 1/500.

**Size-exclusion chromatography**. Size-exclusion chromatography was performed at a flow rate of 0.6 ml/min on a Superdex 200 Increase HR 10/30 column equilibrated in either PBS, or 20 mM Hepes pH 7.5, 130 mM NaCl, or 20 mM Hepes pH 8.2, 130 mM NaCl, as indicated. The column was calibrated with 100 μl of a mixture of ferritin (440 kDa) at 140 μg/ml and thyroglobulin (669 kDa), aldolase (158 kDa), ribonuclease A (13.7 kDa) (all from GE Healthcare), bovine serum albumin (67 kDa), ovalbumin (43 kDa), carbonic anhydrase (29 kDa), and aprotinin (6.5 kDa) (all from Sigma-Aldrich) at 1.4 mg/ml in PBS.

**Western blot and quantification of BAFF**. SDS-PAGE and western blot (using Buffy-2 at 0.5 μg/ml for human BAFF, 852 at 0.5 μg/ml for mouse BAFF and a peroxydase-coupled donkey anti-human IgG for belimumab) were performed according to standard procedures, using WesternBright ECL spray for detection (Advansta) (Supplementary Fig. 9). For the detection of BAFF in fractions of size-exclusion chromatography, proteins were precipitated for 10 min on ice with 5% trichloroacetic acid and 300 μl equivalent of fraction was used for western blot. Coomassie blue staining was performed with a semidry iD Stain System (Eurogentech). WT forms of purified FLAG-mBAFF and FLAG-hBAFF were quantified by SDS-PAGE and Coomassie blue staining against a standard curve of His-BAFF 60-mer[41]. They were used as standards to quantify FLAG-tagged BAFF 3-mers in Superdex-200 fractions (fraction 13 + 14 for mBAFF and 15 + 16 for hBAFF) and naturally cleaved BAFF in concentrated supernatants by western blot using biotinylated anti-FLAG M2 antibody (0.5 μg/ml) followed by IRDye 800CW-coupled streptavidin (100 ng/ml in PBS, 0.5% Tween-20, 1% powdered milk), or Buffy-2 (0.5 μg/ml) followed by IRDye anti-rat (100 ng/ml) or 852 (0.5 μg/ml) followed by IRDye anti-goat (100 ng/ml), and detection with a LI-COR Odyssey infrared fluorescence detector (LI-COR Biosciences).

**Cytotoxic assays with BAFFR:Fas reporter cells**. Untagged BAFF in fractions of the size elution column were tested at the indicated dilutions. 3-mer fractions of FLAG-tagged BAFF were tested at the indicated concentrations, in the presence of a fixed concentration of 1 μg/ml of cross-linking (anti-FLAG, 5A8, Sandy-5) or control (EctoD1) antibodies. Fc-BAFF and BAFF 60-mer were tested at the indicated concentrations. When inhibitors were present, they were added at the indicated concentrations, and unless stated otherwise, pre-incubated for several minutes with ligands (in a final volume of 50 μl medium) before addition of 50 μl of reporter cells (20,000–50,000 cells/well). Reporter cells were incubated with BAFF for 16 h, then assayed for viability by the addition of 20 μl of a solution of phenazine methosulfate at 0.9 mg/ml in PBS and 3-(4,5-dimethylthiazol-2-yl)-5-(3-carboxymethoxyphenyl)-2-(4-sulfophenyl)-2H-tetrazolium at 2 mg/ml in PBS (1:20, v/v) (PMS/MTS), followed by further incubation at 37 °C until adequate color development (2–8 h). Absorbance was measured at 490 nm[44].

**Receptor–ligand interaction ELISA.** For the direct-binding ELISA, BAFFR-Fc, BCMA-Fc, or TACI-Fc (human or mouse) were adsorbed at 1 µg/ml in PBS overnight at room temperature into 96-well immunoplates. The following steps were then performed, with incubations at 37 °C: blocking (in PBS, 4% powdered skimmed milk, 0.5% Tween-20), washing (in PBS, 0.05% Tween-20), incubation with FLAG-tagged BAFF 3-mers of the corresponding species at the indicated concentrations for 1 h (in incubation buffer: PBS, 0.4% milk, 0.05% Tween-20), washing, incubation with biotinylated anti-FLAG at 0.5 µg/ml in incubation buffer for 1 h, washing, incubation with horseradish-coupled streptavidin at 1/4000 dilution in incubation buffer, washing, incubation with 100 µg/ml of 3,3′,5,5′-tetramethylbenzidine in 50 mM citrate-phosphate pH 5, 10% DMSO, 0.03% perborate until color development, addition of 1/3 volume of 2M $H_2SO_4$, and absorbance reading at 405 nm[44]. For the competitive ELISA, titrated amounts of naturally cleaved (untagged) wild-type BAFF were added to adsorbed BAFFR-Fc for 30 min, followed without intermediate washing steps by a fixed and non-saturating concentration of FLAG-tagged BAFF 3-mers (10 ng/ml for human and 50 ng/ml for mouse). Detection with biotinylated anti-FLAG was performed as described for the direct-binding ELISA.

**AlphaLISA.** BAFF in mouse sera was detected by AlphaLISA in white 384-well plates[41]. 5 µl of serum diluted 1:10 in assay buffer (PerkinElmer Life Sciences) was added to 20 µl of a mix of biotinylated Sandy-2 antibody at 75 ng/ml and 0.5 µg of 5A8-coupled acceptor beads in assay buffer and incubated for 1 h at room temperature. Streptavidin-coupled donor beads (1 µg) in 25 µl assay buffer was added. 15 min later, emission at 615 nm after excitation at 680 nm was measured with an Enspire plate reader (PerkinElmer Life Sciences).

hTACI-Fc (100 µg) was mixed to 1 mg of AlphaLISA acceptor beads in 400 µl of 27 mM Na-phosphate pH 8, 5 mM $NaBH_3CN$, 0.03% Tween 20 and incubated for 24 h at 37 °C with agitation, after which time coupling was stopped by addition of 100 µl of carboxymethoxylamine at 65 mg/ml in 0.8 M NaOH for 1 h at 37 °C. Beads were washed twice in 0.1 M Tris–HCl, pH 8 and stored at 4 °C at a concentration of 5 mg/ml in PBS 0.05% Proclin-300. Measures of receptor binding-competent BAFF were performed by mixing 5 µl of mouse serum with 1.5 µg of hTACI-Fc acceptor beads in assay buffer for 3 h at room temperature. Beads were harvested for 10 min at 8000 rpm (6200 × g), washed with assay buffer, suspended in 25 µl of 0.45 µg/ml biotinylated Sandy-2, transferred in white 384-well assay plates, incubated for 1 h at room temperature, after which time 1.5 µg of streptavidin-coupled donor beads in 25 µl of assay buffer was added. Emission at 615 nm after excitation at 680 nm was recorded with an Enspire plate reader (Perkin Elmer).

**Mice treatments.** For B cell stimulation experiments, Sandy-5, 5A8, or EctoD1 (control) antibodies were administered i.p. to $Baff^{E247K/E247K}$ mice at 2 mg/kg on days 0, 7, 14, 21, 28, and 35. Mice were killed by $CO_2$ exposure at day 42 for the analysis of spleens and lymph nodes (inguinal, axillary, and brachial).

**Flow cytometry.** Secondary lymphoid organs were homogenized. Red blood cells were lysed for 5 min on ice in 150 mM $NH_4Cl$, 10 mM $NaHCO_3$, 100 µM $Na_2$-EDTA. Lymphocytes were washed in PBS 2% FCS and filtered on a nylon mesh. Cells were incubated with antiCD16/32 (Fc-block, clone 93, 1/100, eBiosciences) and stained with a mix of anti-CD19-PE.Cy7 (clone eBio1D3, 1/200, eBiosciences), anti-CD93-biot (mAb493, 1/100, prepared in-house by Antonius Rolink), anti-CD1d-FITC (clone 1B1, 1/100, eBiosciences), and anti-CD3-APC (clone 17A2, 1/100, eBiosciences) for 20 min at 4 °C followed by PE-Cy5.5-coupled streptavidin (1/200) and analysis with a FACS Canto (BD Biosciences). Data were analyzed with the FlowJo software.

**Mouse B splenocyte survival assay.** Mouse splenocytes were purified with an EasySep™ mouse isolation kit (StemCell Technologies, #19854). Purified B splenocytes were labeled for 8 min at 37 °C in PBS, 1% FCS, 2 µM carboxyfluorescein succinimidylester (CFSE) and cultured in round-bottomed 96-well culture plates, at a density of $1.25 \times 10^6$ cells/ml in 200 µl of medium (RPMI, 10% FCS, 50 µM of 2-mercaptoethanol, 100 U/ml of penicillin, and 100 µg/ml of streptomycin medium) with titrated amounts of size-fractionated FLAG-mBAFF trimer in the presence or absence of 1 µg/ml of cross-linking (5A8, Sandy-2) or control (EctoD1) antibodies. After 72 h, cells were stained with 100 ng of propidium iodide per sample and analyzed by flow cytometry using an AccuriC6 flow cytometer (BD Bioscience). The percentage of viable cells was calculated as the number of live cells (CFSE+, PI-) divided by the number of cells (CFSE+) × 100[41].

**Crystallography.** Crystals of $His_6$-hBAFF H218A in complex with three Fab fragments of belimumab were obtained by mixing 0.1 µl protein solution (28 mg/ml in 20 mM Hepes pH 7.5, 150 mM NaCl) with 0.1 µl reservoir solution (9% (w/v) PEG4000, 0.1 M $MgCl_2$, 0.1 M HEPES pH 7.5) using the sitting drop vapor diffusion method at 293 K. Before flash cooling in liquid nitrogen, crystals were cryoprotected by reservoir solution supplemented with 20% (v/v) ethylene glycol.

Diffraction data were collected at 100 K at the Swiss Light Source beamlight X06SA (SLS, Villigen, Switzerland) using an Eiger X 16 M detector (Dectris, Baden-Daettwil, Switzerland). Data were processed to 2.9 Å resolution using the programs XDS and XSCALE[49]. The crystals belong to space group P 2₁ with a solvent content

of 64% (Matthews coefficient 3.45). They contain 2 hexameric complexes per asymmetric unit. Each complex, consisting of a BAFF 3-mer bound by 3 Fab fragments of belimumab, has a 3-fold non-crystallographic symmetry. The phase information necessary to determine and analyze the structure of the BAFF - belimumab Fab complex was obtained by molecular replacement using the program PHASER[50] and the published structures of BAFF 3-mer (PBD-ID 1KD7) and of an Fab fragment (PDB-ID 7FAB). Subsequent model building and refinement was performed with the software packages CCP4 and COOT[51,52]. For the measure of the free R-factor, a measure to cross-validate correctness of the final model, about 0.6% of measured reflections were excluded from the refinement procedure (Supplementary Table 4). Several rounds of manual model building in COOT and bulk solvent correction, positional, B-factor and TLS refinement using REFMAC yielded the final model[51,52]. The Ramachandran Plot of the final model shows 89.4% of all residues in the most favored region, 10.0% in the additionally allowed region, and 0.5% in the generously allowed region. The residues Asp153 (E), Asp153(I), and Asp153(N) of the Fab fragments are found in the disallowed region of the Ramachandran plot. Data collection and model statistics are shown in Supplementary Table 4. The atomic coordinates and structure factors have been deposited in the Protein Data Bank, www.pdb.org (PDB-ID code 6ERX). Buried surface areas were calculated with AreaIMol[53]. Images were generated with the PyMOL Molecular Graphics System, Schrödinger, LLC.

**Electron microscopy data acquisition and image processing.** Negatively stained specimens were prepared following an established protocol with minor modifications[54]. Specifically, 2.5 µl of sample was applied to glow-discharged copper electron microscopy (EM) grids covered with a thin layer of continuous carbon film, and the grids were stained with 2% (w/v) uranyl formate. The sample concentrations of BAFF 60-mer, BAFF 60-mer bound with atacicept, and BAFF 3-mer bound with Fab were 23, 20, and 8.8 µg/ml, respectively. These grids were imaged on a Tecnai T12 electron microscope (FEI) operated at 120 kV at a nominal magnification of 67,000× using a 4k × 4k CCD camera (UltraScan 4000, Gatan), corresponding to a calibrated pixel size of 1.68 Å on the specimen level.

The EM data were processed with SAMUEL and SamViewer[55]. Negative-stain EM images were binned over 2 × 2 pixels for further processing, yielding a pixel size of 3.36 Å. Particle picking was performed using a semi-automated procedure[56]. 2D classification of selected particle images was carried out with 'samclasscas.py', which uses SPIDER operations to run 10 cycles of correspondence analysis, K-means classification, and multi-reference alignment[57].

**Statistics.** Group sizes for animal experiments were chosen to detect 40% differences between conditions, with variation coefficients of 15–20% (4 animals/group). Animals were assigned randomly to groups, except for males and females that were distributed equally between groups. Analyses were not blinded. Normal distribution of data was confirmed by normality tests using Prism (D'Agostino and Pearson test for $n \geq 8$, Kolmogorov–Smirnov test for $n = 6$ or 7), or was assumed for smaller size groups. One-way analysis of variance with Bonferroni's multiple comparison tests was used to compare selected groups using Prism. One-way analysis of variance assumes that the standard deviation of the different groups is equal, an assumption that was not always met according to Bartlett's test for equality of variance. But as this assumption is of little importance when group sizes are (almost) equal, this was ignored.

**Data availability.** The authors declare that the data supporting the findings of this study are available within the article, in its supplementary information files, as a dataset[58], or are available upon reasonable requests to the authors. The structural datasets are available in the Protein Data Bank repository, www.pdb.org, under the PDB-ID code 6ERX.

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

## Acknowledgements

We are grateful to Susan Kalled and Linda Burkly (Biogen, Boston) for providing BAFF$^{-/-}$ mice, and to Anne-Marie Mérillat (University of Lausanne) for assistance with creation of knock-in mice. This work was supported by grants from the Swiss National Science Foundation (to P.S.), and by a research grant from EMD Serono, a subsidiary of Merck, KGaA (to P.S.).

## Author contributions

P.S. conceived the experiments and generated protein complexes for EM and crystal-lography. D.D., P.S., and E.H. generated knock-in mice. M.G.C. and M.L. performed and analyzed EM. K.M. and A.L. performed crystallography. M.V. performed most other experiments, with the help of L.W., A.T., D.C., C.K.-Q., S.S.-M., C.R.S., M.E., and P.S. A.R., E.S., C.R.S., F.M., X.J., and H.H. contributed experimental ideas and key reagents to perform them. P.S. wrote the paper with the help of M.V. All the authors reviewed the results and approved the final version of the manuscript.

## Additional information

**Competing interests:** A.L. and M.K. are employees of Proteros biostructures GmbH. E. S., Y.F.N., and X.J. are employees of EMD Serono. H.H. is employee of Merck, KGaA. The remaining authors declare no competing interests.

