## [Peer Review File · Nature Communications]

Reviewers' comments:

Reviewer #1 (BAFF/APRIL, auto-antibody)(Remarks to the Author):

In this manuscript, the authors convincingly demonstrate that the flap region of BAFF is not only important for generation of BAFF 60-mers (as previously shown by multiple laboratories) but is also important for the biological activity of BAFF (both in vitro and in vivo).

Minor comments:

1. Page 7, second paragraph: I believe the authors are referring to Fig 2A, rather than Fig 3A.
2. Page 9, last paragraph: BAFF H242A has a mutated flap region, yet it remains biologically active. Thus, the statement "the flap is required for the activity of BAFF" needs to be modified.

Reviewer #2 (X-ray, antibody-antigen complexes)(Remarks to the Author):

The manuscript describes the results from numerous studies of BAFF and its interactions with receptors, including crystal and EM structures of BAFF in complex with the inhibitory antibody Belimumab and EM studies of BAFF in complex with atacicept. The combined results help to elucidate the function of the BAFF flap region. The study will be of high interest for those interested in BAFF and related cytokines and for SLE therapeutics.

Throughout the manuscript there are quite a few grammatical/spelling errors. Please have the paper edited carefully for these problems.

Does the BAFF H218A mutation affect its affinity for Belimumab? From the figures it is hard to see whether this mutation falls in the epitope- please comment on this. Also, in the section describing the crystal structure, please include more details to describe the complex, such as the size of the interface, numbers of contacts, which residues are in the epitope, etc. In Fig. 8abc, there are some side chains shown for the Fab and the BAFF, but the figure legend doesn't say what these are, please include those details.

Figure 1, panels B, In figure legend please state that the B panel is a gel of human Baff.

Fig. 1, D legend and elsewhere, instead of anti-FLAG 'revelation': please use 'detection' in place of 'revelation'.

Fig. 4B, the legend does not say what the colors on the residues represents. Please explain.

p. 25, Fab digestions- did you really digest for 100 hours?

p. 28, Crystallography section, change 'flash freezing' to 'flash cooling'. Please include the Matthews coefficient and solvent content. The unit cell dimensions in Table S5, and the presence of 6 complexes in the asymmetric unit suggest at least pseudo cubic symmetry, i.e. space group P2(1)3 – please double check that the space group assignment is correct. If P21 is the correct space group, please comment briefly in the methods section about the non-crystallographic symmetry.

We thank both reviewers for their appreciation of the manuscript and for their helpful comments. Changes made in the manuscript and the supplemental information in response to reviewer's comments are highlighted in yellow, grammatical changes requested by reviewer 2 that do not change the scientific content are highlighted in blue, and changes made to match editorial policy (maximal number of characters in subtitles, description of statistics, etc...) are highlighted in green.

Reviewer #1

In this manuscript, the authors convincingly demonstrate that the flap region of BAFF is not only important for generation of BAFF 60-mers (as previously shown by multiple laboratories) but is also important for the biological activity of BAFF (both in vitro and in vivo).

Minor comments: □

1. Page 7, second paragraph: I believe the authors are referring to Fig 2A, rather than Fig 3A.

Answer: We thank the reviewer for having spotted this mistake that we have corrected.

2. Page 9, last paragraph: BAFF H242A has a mutated flap region, yet it remains biologically active. Thus, the statement "the flap is required for the activity of BAFF" needs to be modified.

Answer: The reviewer is right, BAFF with the mutation H242A (or H218A in human BAFF) remains biologically active, at least on reporter cells. However, this result should be interpreted as "residue H242 is not required for the activity of BAFF" and not as "the flap is not required for the activity of BAFF". Indeed, the activity of BAFF can be disrupted by another mutation in the flap, E223K (E247K in mouse BAFF). We thus believe (and show) that the flap is required for the activity of BAFF, and we have maintained the statement as initially written. Figure 4A shows the differential importance of different amino acids in the flap region of BAFF: glutamic acid 223 is essential for the interaction of two flaps, and its mutation to lysine strongly impairs the activity of BAFF. Histidine 218 does not seem to impair the interaction between two flaps, and therefore does not abolish activity, but it is required for the formation of BAFF 60-mers (whose activity is superior to that of BAFF 3-mers).

While addressing this reviewer's comment, we noticed an error in panel 4D, where numbers for the human mutations were given for the mouse proteins. This has now been corrected.

Reviewer #2

The manuscript describes the results from numerous studies of BAFF and its interactions with receptors, including crystal and EM structures of BAFF in complex with the inhibitory antibody Belimumab and EM studies of BAFF in complex with atacept. The combined results help to elucidate the function of the BAFF flap region. The study will be of high interest for those interested in BAFF and related cytokines and for SLE therapeutics. □ □ Throughout the manuscript there are quite a few grammatical/spelling errors. Please have the paper edited carefully for these problems.

Answer: A native English speaker has now edited the manuscript; we hope that we have improved grammar and spelling.

Does the BAFF H218A mutation affect its affinity for Belimumab? From the figures it is hard to see whether this mutation falls in the epitope- please comment on this.
Answer: As shown in the figure below, the mutation H218A is not part of the binding site of belimumab:

Fig. for the reviewer only. Mutation H218A in the flap of BAFF is not part of the binding site of belimumab.

Given that mutation H218A is not part of the epitope, one would not expect a major change in affinity. We have addressed this qualitatively in a competition ELISA, where the biotinylated (monomeric) Fab of belimumab was given the choice to bind BAFF (WT or H218A) coated in the ELISA plate or soluble BAFF (WT or H218A) added as competitors. In these experiments, BAFF H218A was a competitor as efficient as BAFF WT, indicating that the mutation does not grossly affect affinity for belimumab. For this experiment, we have used Fc-BAFF (that cannot form 60-mer) instead of His-BAFF (as used for crystallography), because WT His-BAFF forms 60-mers that are not recognized by belimumab.

We have edited a sentence in the manuscript (page 16):

“To prevent association of BAFF 3-mer into 60-mer, BAFF contained mutation H218A into the flap region²³” → “To prevent association of BAFF 3-mer into 60-mer, mutation H218A was introduced into the flap region²³, which did not grossly affect the affinity of belimumab for BAFF (Supplementary Fig. 6).”

A supplementary figure (Fig. S6) was added in the supplemental information. It is copied below:

Supplementary Figure 6 (related to Figure 8). Mutation H218A in the flap of BAFF does not grossly alter the affinity for belimumab.

The fusion proteins Fc-BAFF, with or without the point mutation H218A in the flap of BAFF, were expressed in HEK 293T cells by transient transfection, then affinity purified on immobilized TACI-Fc. Fusion of a Fc moiety at the N-terminus of BAFF has at least two consequences: i) BAFF cannot form 60-mers for steric hindrance reasons and thus is recognized by belimumab (see Fig. 6A) and ii) fusion of a dimeric Fc to a trimeric ligand leads to the formation of active hexamers with two trimeric ligands intrinsically cross-linked by three dimeric Fc¹. Fc-BAFF should thus be active regardless of mutations in the flap, and the flap should be accessible to belimumab.

A.- Coomassie blue staining of 10 μ g of Fc-BAFF WT and Fc-BAFF H218A produced by transient transfection in HEK 293T cells and affinity purified on immobilized TACI-Fc.

B.- Fc-BAFF WT and H218A were incubated at the indicated concentrations for 16 h with BAFFR:Fas reporter cells, after which time cell viability was measured with the PMS/MTS cell viability assay. The experiment was performed twice.

C.- Fc-BAFF WT or Fc-BAFF H218A were coated in an ELISA plate. The biotinylated Fab fragment of belimumab was added at various dilutions, and binding to the coated Fc-ligand was detected with peroxidase-coupled streptavidin. The arrow indicates the concentration chosen for subsequent competition assays (see panels D and E). Mean \pm SEM of duplicates.

D.- Fc-BAFF WT was coated in an ELISA plate. After the blocking step, soluble Fc-BAFF WT or H218A was added in wells at twice the indicated final concentrations. Without wash, biotinylated belimumab Fab was subsequently added and mixed immediately to reach a fixed final concentration of 50 ng/ml. After incubation and a washing step, bound Fab was detected as described in panel C. Mean \pm SEM of duplicates.

E.- Same as panel D, except that Fc-BAFF H218A was coated in the ELISA plate. Mean \pm SEM of duplicates. The experiment in panels C-E was performed twice

1. Holler N, *et al.* Two adjacent trimeric Fas ligands are required for Fas signaling and formation of a death-inducing signaling complex. *Mol Cell Biol* **23**, 1428-1440 (2003).

Also, in the section describing the crystal structure, please include more details to describe the complex, such as the size of the interface, numbers of contacts, which residues are in the epitope, etc. In Fig. 8abc, there are some side chains shown for the Fab and the BAFF, but the figure legend doesn't say what these are, please include those details.

Answer: we agree with the reviewer that we did not adequately describe the binding site. We have modified the text to mention the size of the interface:

“A closer look at the structure of the BAFF - belimumab Fab complex showed that both heavy and light chains of belimumab contacted BAFF over its entire height, including the receptor-binding site and the flap regions (Fig. 8A).” → “A closer look at the structure of the BAFF - belimumab Fab complex showed that both heavy and light chains of belimumab contacted BAFF over its entire height, forming two distinct interfaces with the receptor-binding site (664 Å²) and the flap region (333 Å²), for a total surface area of 995 Å² (963 to 1014 Å² for the 6 copies of the asymmetric unit) (Fig. 8A).”

We have also included a supplementary figure describing the binding interface. Fig. S7 includes a list of residues in the interaction site, and a visual representation of the interface showing most of these residues. Fig S7 is copied below:

Supplementary Figure 7 (related to Figure 8). Interface of belimumab bound to BAFF.
 A. Residues of BAFF that are closer than 4 Å from those of belimumab are connected by black lines. CDR: complementarity-determining regions. DXL: Asp-Xxx-Leu motif. The DXL motif is at the center of the binding of BAFF with its receptors (Fig. 8B, E).
 B. Detailed view of the interaction site between belimumab and BAFF. Most residues at the contact site are identified.

Figure 1, panels B, In figure legend please state that the B panel is a gel of human Baff.

Answer: this has been done.

□

Fig. 1, D legend and elsewhere, instead of anti-FLAG ‘revelation’: please use ‘detection’ in place of ‘revelation’.

Answer: we have changed this throughout the manuscript.

□

Fig. 4B, the legend does not say what the colors on the residues represents. Please explain.

Answer: colors have been changed according to recommendations of the journal not to use red and green for color-blind readers. We have now added in the legend a description of the color (magenta= acidic, blue = basic, turquoise=tyrosine that pairs with H218).

p. 25, Fab digestions- did you really digest for 100 hours?

Answer: yes, the Fab was generated in a 4 days digestion. To prevent confusion, we have now written 4 days instead of 100 h.

p. 28, Crystallography section, change 'flash freezing' to 'flash cooling'.

Answer: this was done.

Please include the Matthews coefficient and solvent content.

Answer: we have added this on page 28: "The crystals belong to space group P 2₁ with a solvent content of 64% (Matthews coefficient 3.45). They contain 2 hexameric complexes per asymmetric unit."

The unit cell dimensions in Table S5, and the presence of 6 complexes in the asymmetric unit suggest at least pseudo cubic symmetry, i.e. space group P2(1)3 – please double check that the space group assignment is correct.

Answer: there are 2 hexameric complexes in the asymmetric unit. We have double-checked the indexing of the crystals and saw no hint for an additional crystallographic 3-fold axis. The space group is indeed P2(1).

If P21 is the correct space group, please comment briefly in the methods section about the non-crystallographic symmetry.

Answer: the following sentence has been added on page 28: "Each complex, consisting of a BAFF 3-mer bound by 3 Fab fragments of belimumab, shows a 3-fold non-crystallographic symmetry".

REVIEWERS' COMMENTS:

Reviewer #1 (Remarks to the Author):

My previous minor concerns have been adequately addressed.